



# Modelling the impact of anthropogenic aerosols on the CCN concentrations in rural boreal forest environment

Petri Clusius[1], Metin Baykara[2], Carlton Xavier[3,4], Putian Zhou[1], Juniper Tyree[1], Benjamin Foreback[1,6], Mikko Äijälä[6,1], Frans Graeffe[1], Tuukka Petäjä[1], Pauli Paasonen[1], Markku Kulmala[1], Paul I. Palmer[5], Michael Boy[1,6,7]

[1]Institute for Atmospheric and Earth Systems Research, University of Helsinki, P.O. Box 64, 00014 Helsinki, Finland
[2]Climate and Marine Sciences Department, Eurasia Institute of Earth Sciences, Istanbul Technical University, Maslak, Istanbul, 34469, Turkey
[3]Department of Physics, Lund University, SE-22100 Lund, Sweden
[4]Research Department, Swedish Meteorological and Hydrological Institute, SE-60176 Norrköping, Sweden
[5]School of GeoSciences, University of Edinburgh, Edinburgh, UK
[6]Atmospheric Modelling Centre – Lahti, Lahti University Campus, Lahti, Finland
[7]School of Energy Systems, Lappeenranta-Lahti University of Technology, Lappeenranta, 53850, Finland

*Correspondence to*: Petri Clusius (petri.clusius@helsinki.fi)

**Abstract.** The radiative properties of the clouds depend partially on the cloud droplet number concentration, which is determined by the available cloud condensation nuclei (CCN) when the clouds are formed. In turn, the CCN concentrations are determined by the atmospheric particle size distribution and their chemical composition. Here, we use a novel modelling framework (SOSAA-FP) that combines a) backward trajectories from the FLEXPART airmass dispersion model, b) a detailed description of atmospheric chemistry and aerosol dynamics from the SOSAA model, and c) global emission datasets to model the formation and lifetime of atmospheric aerosols and their potential to act as CCN at the boreal forest measurement site SMEAR II. This enables us to examine the origin and history of the gas and aerosol components observed at SMEAR II station in southern Finland. In this study, we apply the SOSAA-FP to simulate a period from March to October 2018 with one hour time resolution, focusing on the concentrations of CCN between 0.1–1.2% maximum supersaturation as calculated by the κ-Köhler theory. We find that the model $PM_1$ fraction of primary particles, sulfates and secondary organic aerosol correlate well with the observed organic aerosol and sulfate trends and explain most of the observed organic aerosol and sulfate $PM_1$ mass. Our results show that primary particle emissions play a considerable role in CCN concentrations even at a rural site such as SMEAR II. Atmospheric cluster formation rates had a relatively weak impact on the CCN concentrations in the sensitivity runs. Enhanced nucleation increased (decreased) the CCN concentrations for the highest (lowest) maximum supersaturation. Without any new particle formation the modelled CCN concentrations dropped as much as 36 % for CCN 1.2% and increased by 29% for 0.1% supersaturation, whereas omitting primary particle emissions had a decreasing effect in all calculated CCN supersaturation classes.



## 1 Introduction

The atmospheric energy budget is heavily influenced by water, be it in vapour, liquid or solid phase. Of these, undoubtedly the two latter – clouds – are far more complex when it comes to quantifying their influence on radiative forcing (RF). Clouds affect the path and intensity of downward and upward radiation in visible and thermal wavelengths by absorption and scattering. Broadly, the process of forming cloud droplets requires two ingredients: first, excess moisture – above 100% relative humidity, i.e. supersaturation state – and second the presence of cloud condensation nuclei (CCN), which act as condensation seeds for heterogeneous nucleation of water. The first condition is usually met where moist air is being lifted by convection, and the adiabatic cooling of the air brings the vapour below the dew point. Depending on the updraft velocity, which roughly determines the maximum achievable supersaturation, the critical diameter of the CCN, i.e. particle size above which over 50% of particles activate as CCN, is commonly found between 40–100 nm (Dusek et al., 2006), or even below ( Svensmark et al., 2024).

CCN can be originate from primary particles, that is, particles that are directly emitted to the atmosphere (and possibly processed further in the atmosphere), or by secondary particles via new particle formation (NPF), where the particles are formed through clustering and condensational growth of atmospheric vapours. These two sources are separate only in the most ideal sense; in the atmosphere the particles are mixed due to coagulation, and secondary aerosol mass is forming on primary particles. Consequently our definition that demarcates the two processes are not clearly defined, and remain difficult to determine with measurements alone (Kerminen et al., 2012). The origins of CCN are relevant to future climate predictions as it is likely that the anthropogenic emissions of both primarily emitted particles as well as the precursor gases involved in NPF is changing with emission mitigation policies, and will likely change further in the coming years due to the transition of fossil fuel based energy production and transport to renewables and nuclear energy.

NPF is estimated to contribute to a significant fraction of the global CCN. Several model studies have shown that globally, about half of all CCN are secondary organic aerosol particles originating from NPF (Merikanto et al., 2009; Yu and Luo, 2009; Gordon et al., 2017), while other model studies suggest that more modest effect (Pierce and Adams, 2009; Reddington et al., 2011). Gordon et al. (2017) estimated NPF to account for 54% of the global CCN at 0.2% supersaturation in the present day. Pierce et al. (2014) made similar estimations for particles between 50–100 nm diameter in a 1 year size distribution measurements at Egbert, Canada. Estimates vary still markedly depending on the location, anthropogenic influence and size class, and in general modelling the effect of NPF to CCN remains difficult (Ren et al., 2021).

Considerable work has been done to understand cluster formation in different conditions and geographical locations (e.g. Hirsikko et al., 2011; Kirkby et al., 2011; Almeida et al., 2013; Riccobono et al., 2014; Dunne et al., 2016; Chamba et al., 2023; Zhao et al., 2024). The starting point of NPF, cluster formation, is known to occur through clustering of sulfuric acid and basic compounds such as ammonia or amines (Sipilä et al., 2010), but other pathways, involving for example large organic molecules or iodic acids are been observed in different environments and laboratory experiments (Elm et al., 2020) Zhao et al., 2024). When discussing CCN, equally important part to the rate of cluster formation are their growth rates, as the





clusters are quickly scavenged by larger particles. The growth of the sub-10nm clusters is believed to come largely through condensation of (usually highly oxygenated) extremely low volatility organic vapours and sulfuric acid (Ehn et al., 2014). This makes the VOC emissions, their oxidation, and partitioning within the aerosol size distribution crucial to the atmospheric formation of CCN. The most abundant emitted biogenic volatile organic compounds (BVOC) isoprene, mono- and sesquiterpenes are know to produce upon their oxidation a vast spectrum of low volatility vapours, many of which can con-

dense on the smallest particle sizes, or even participate in cluster formation (Dada et al., 2023). Modelling of particle growth via SOA is usually based on the saturation vapour pressure of the oxidation products, or parametrised in terms of mass yield, as is often done in global models. Regarding anthropogenic VOC emissions (AVOC), most models lack a mechanistic understanding of their autoxidation and the resulting components, contributing to the uncertainty in estimating human influence on SOA formation and thereby CCN concentrations in urban areas. Steps towards closing this knowledge gap have been

taken in the recent years (Garmash et al., 2020; Wang et al., 2021; Pichelstorfer et al., 2024). There is also evidence on heterogeneous chemistry between VOC oxidation products and aerosols, affecting secondary organic aerosol (SOA) yields, but the global implications of these processes are still unclear.

Studies have linked BVOC emissions to CCN concentrations (e.g. Riipinen et al., 2011; Petäjä et al., 2021; Yli-Juuti et al., 2021), but due to the intertwined nature of primary and secondary aerosols, it is not clear how much of this link translates to

the contribution of cluster formation specifically and NPF in general. Atmospheric chemical composition and aerosol processes in a particular location is generally affected by the airmass history, especially for components with long atmospheric lifetimes and processes with long timescales. While CCN concentrations are linked to high SOA mass and increased temperatures along trajectories, from local observations it is not clear what are the underlying independent processes during the upstream history of the airmasses. If local observations of aerosol processes are considered regional, wrong conclusions of the

dynamics will be drawn. E.g. Hakala et al. (2023) showed that the apparent shrinkage of aerosol population at Hada al Sham, Saudi-Arabia, was simply due to development of air mass transport in a manner that the particles arriving to the observation site during the afternoon/evening had grown less than those arriving before them, due to spatial and temporal heterogeneities in concentrations of vapours causing the growth. Studies at SMEAR II (Station for Measuring Ecosystem-Atmosphere Relations) station at Hyytiälä, Finland, have linked local NPF events to airmass trajectories arriving from the north-westerly or

north-easterly direction over the North Atlantic or Arctic Ocean, consistent with conditions such as low condensation sink, ample short wave radiation and availability of sulfuric acid (Riuttanen et al., 2013) Lubna, Tuomo and others]. These conditions are somewhat different from those when high SOA loadings and increased temperatures along trajectory are observed. This implies that cluster formation and CCN formation are in general decoupled, and a favourable sequence of events is necessary for cluster formation to lead to CCN. Such would be the case of marine air arriving and advecting over forested area,

eventually leading to increased CCN concentrations. However, the difficulty of separating primary particle emissions from secondary aerosol formation remains, as observation-based models aiming to describe measured aerosol types often struggle to decisively classify aerosols as secondary or primary, and need very detailed chemical information to succeed (Zhang et al., 2009).



In the attempt of separating secondary aerosol formation, primary emissions (and other anthropogenic influence), seasonal
variation of the BVOC emissions and variability of the origins of the airmasses, and quantifying their respective contribu-
tions, we have applied a one-dimensional column model in a Lagrangian mode. This allows us to attribute and quantify how
emissions and atmospheric processes have affected the aerosol population before it is eventually observed on a measurement
station. In this study, we have developed a modelling framework which can utilize detailed aerosol process models on the
necessary spatial scale to describe the airmass history at least one week backwards. This modelling framework was used to
address the following questions: 1) how much of the given CCN number concentration or composition can be attributed to
primary or secondary sources; 2) how sensitive is the CCN number concentration to changes in emissions of some key in-
gredients and cluster formation rates; 3) given possible future changes in the tested parameters, what can we expect to hap-
pen to CCN number concentrations. We concentrate here on three components that affect CCN concentration: cluster forma-
tion rates, anthropogenic primary number emissions and BVOC emissions. These are referred from here on as NUC, PNE
and BIO, respectively. For the benefit of the reader, Table 1 describes the abbreviations frequently used in this paper.

*Table 1: Frequently used abbreviations*

| | |
|---|---|
| BIO | BVOC emissions in the model |
| NUC | Cluster formation rates in the model |
| PNE | Anthropogenic $PN_{1\mu m}$ particle number emissions in the model |
| $f_I$ | factor of change in BIO, NUC or PNE |
| $f_O$ | factor of change in some model output |
| $R = R(f_I, f_O)$ | Model response to change in BIO, NUC or PNE, calculated from $f_I$ and $f_O$ |
| (A/B)VOC | (Anthropogenic/Biogenic) volatile organic compound |
| NPF | New particle formation, including formation of clusters and their growth |
| HOM | Highly oxygenated molecules, C:O ratio < 0.7 (Bianchi et al., 2019) |
| $CCN_X$ | Cloud condensation nuclei at supersaturation x % |
| [C] | brackets refer to concentration of $C$ (cm⁻³ if not stated otherwise) |
| PSD | particle size distribution |

## 2 Methods and data

The following sections describe the SOSAA-FLEXPART (later, SOSAA-FP) modelling framework, consisting of SOSAA
(model to Simulate the concentrations of Organic vapours, Sulphuric Acid and Aerosols, (e.g. Zhou et al., 2017; Chen et al.,
2021; Boy et al., 2022) and FLEXPART (FLEXible PARTicle dispersion model, Stohl et al., 1998; Stohl and Thomson,
1999; Stohl et al., 2005; Pisso et al., 2019) models, and its input data. In addition to a general overview, the individual steps



necessary to use the SOSAA column model are described. We also include a description of the measurement data used to evaluate the model. For brevity, some details are found in the Supplementary material.

## 2.1 SOSAA-FLEXPART model framework (SOSAA-FP)

The SOSAA-FP framework simulates the effects of emissions, atmospheric chemistry, physics and meteorology on particle number size distribution, composition and gaseous compound concentrations during the long-distance transport to a chosen point location. SOSAA-FP uses global, gridded emission, concentration and meteorological data and is therefore not dependent on the availability of measurements at the location. However, as comparing a model to observations is beneficial in evaluating its performance, we chose SMEAR II as the end point of the trajectories.

SOSAA-FP consists of a 1-dimensional column model SOSAA, that simulates atmospheric chemistry and aerosol physics along a FLEXPART trajectory. Nominally, the model follows a mean airmass trajectory, which is calculated in advance using FLEXPART in backward mode. During the SOSAA simulation, the model is constrained by meteorological conditions (we used ERA5: Fifth generation of ECMWF atmospheric reanalyses of the global climate dataset, Hersbach et al. (2018); Hersbach et al. (2020)) at the mean location of the transported air parcels. The concentrations of all gas phase compounds are calculated by the SOSAA chemistry module based on the emissions (and meteorology) along the trajectory, except for $[O_3]$ that was read in throughout the simulation from the CAMS global atmospheric composition dataset. In addition, $[CO]$ and $[SO_2]$ were initialized using CAMS data. Similarly, the aerosol size distribution, concentrations and composition are calculated by the aerosol module. The chemistry and aerosol modules are updated with the mean emissions at any given location. These means were calculated using FLEXPART's Source Receptor Relationship output as weighting factors, and are discussed in more detail in section 2.1.2. The losses of the gases and particles are modelled with a simplified dry deposition to the ground and vegetation, further discussed in section 2.1.3. The SOSAA-FP framework is schematically presented in Fig 1. Each simulated trajectory contains the history of the airmass arriving at the station at a given time, shown in Fig 1 with the red frames. When the procedure is repeated for a set of trajectories, a time series at the station is obtained, represented in the schematics by the bottom surface plot showing the time series of the vertical profile of HOM monomer concentrations.

The SOSAA-FP framework involves many pre- and post-processing steps before the SOSAA model can be run. We have made these processes easier with a suite of processing scripts, which streamlines the data processing on an HPC computer. Similarly to the ARCA-box (Atmospherically Relevant Chemistry and Aerosol model, Clusius et al., 2022), the SOSAA trajectory model has a graphical user interface, which can be used in setting up the model and data analysis. With this software suite the SOSAA-FP framework can be straightforwardly applied from scratch to model any location on the globe, without the need to acquire site-specific input data for the models.





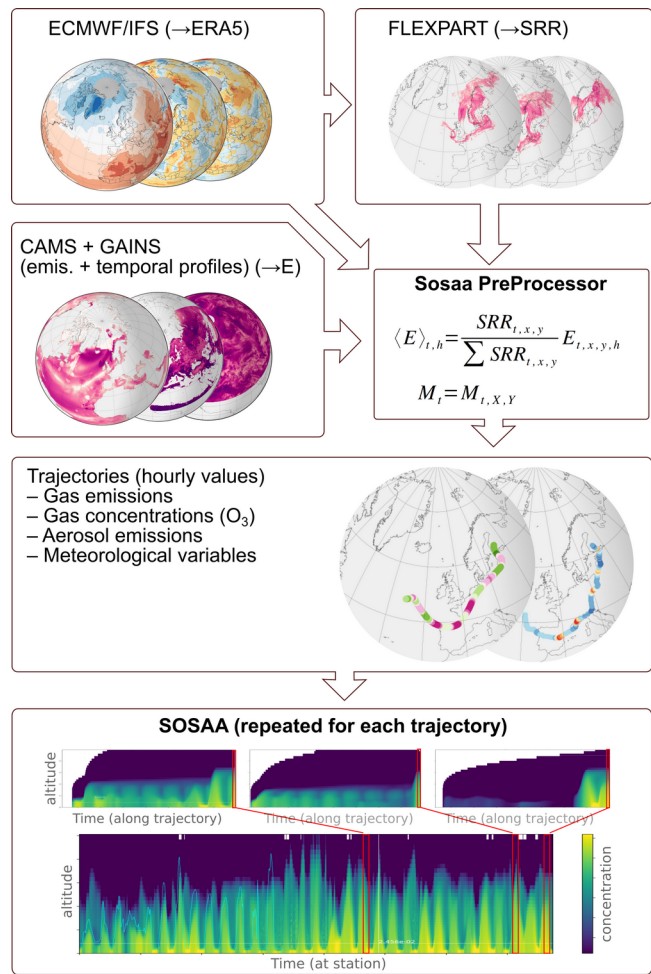

Figure 1: Schematic workflow of the SOSAA-FP framework. The boxes show the main steps taken to arrive with the results presented in this work (bolded headlines show tools which are developed for this work). The gridded data input (inboxes with the globes) is acquired from emission datasets and meteorological reanalysis, or calculated with an airmass dispersion model.

### 2.1.1 Flexpart trajectories

FLEXPART (FLEXible PARTicle dispersion model, Stohl et al., 1998; Stohl and Thomson, 1999; Stohl et al., 2005; Pisso et
al., 2019) is a dispersion model used to trace airmasses both in the forward-time and backward-time mode (Seibert and Frank, 2004). In this study FLEXPART was used in the backward mode. FLEXPART needs gridded meteorological reanalysis data, and in this study, we used ERA5 dataset (Copernicus Climate Change Service, Hersbach et al., 2018), extracted in $0.5° \times 0.5°$ horizontal resolution and 137 (hybrid) vertical layers. FLEXPART starts with a predetermined number of particles (in this study inert and massless tracers), and disperses them forward or backward in time. This leads to a plume of tracers,



generally spreading more the further away the air masses are from the release location. One of FLEXPART's outputs is the Source Receptor Relationship (SRR, in unit of seconds), which describes the time the air parcels are affected by emissions at different locations. When multiplied with the gridded emission fluxes, one gets the concentration at the receptor (the tracer release locations). Each output time (1 hour interval) FLEXPART uses k-means clustering to group the dispersed particles into a few (in our case five) clusters, whose centre of mass locations are saved separately. From these clusters a single mean trajectory line is also calculated. In this study, some of the meteorological data (temperature, pressure, humidity, friction velocity and land-sea mask) was extracted from the single trajectory points from the ERA5 data, whereas the SRR-weighted averages were used for all emissions, short wave radiation, leaf area index and albedo. The hourly mean mixing heights are calculated by FLEXPART using the lowest altitude where the Richardson number exceeds 0.25, and by taking a weighted average of the tracer plume.

### 2.1.2 Emission Inputs

Since the SOSAA model is a 1-dimensional column model which is only run once per trajectory, the 3-dimensional emission and airmass dispersion fields need to be averaged to represent a mean trajectory. In order to calculate the mean emissions at any given time, normalized SRR were used as weighting factors in a pre-processing step (SOSAA PreProcessor, SPP). In this study, SPP is configured to use global emissions developed as part of the Copernicus Atmosphere Monitoring Service (CAMS, Granier et al., 2019); global anthropogenic emissions inventory (CAMS-GLOB-ANT), biogenic global emissions (CAMS-GLOB-BIO), global oceanic emissions (CAMS-GLOB-OCE).

The anthropogenic dataset used for the model simulations was CAMS-GLOB-ANT dataset which is a globally gridded anthropogenic emissions inventory based on emissions provided by Emissions Database for Global Atmospheric Research (EDGAR) developed at Joint Research Center (JRC) (https://edgar.jrc.ec.europa.eu/, last access April 2024, Crippa et al., 2018), and Community Emissions Data System (CEDS) (Hoesly et al., 2018). CAMS-GLOB-ANT dataset has a spatial resolution of $0.1° \times 0.1°$ in latitude and longitude and provides monthly averages of the global emissions of 36 compounds, 25 of which are speciated volatile organic compounds (Huang et al., 2017), including the main air pollutants such as NOx, $NH_3$, $SO_2$, CO and $CH_4$ for 16 emission sectors. These emission sectors are based on GFNR (Gridded Nomenclature For Reporting) sector classification. The methodology of how these emissions were generated for the 2000–2023 period is explained in detail by Soulie et al. (2023). To convert these emissions to hourly data CAMS-GLOB-TEMPO (Guevara et al., 2021) global emission temporal profiles are used. This temporal profile dataset has a spatial resolution of $0.1° \times 0.1°$ in latitude and longitude and includes monthly, weekly (day-of-the-week), daily (day-of-the-year) and diurnal temporal profiles for the main air pollutants (NOx, $SO_2$, NMVOC, $NH_3$, CO and $PM_{2.5}$, here used for number emissions of $PN_{1\mu m}$) and the greenhouse gases ($CO_2$ and $CH_4$). Temporal profiles are mainly for the following sectors; energy industry, refineries, residential combustion, manufacturing industry, road transport, aviation and agriculture.

For the biogenic gaseous emissions, CAMS-GLOB-BIO and CAMS-GLOB-OCE datasets were used. The emissions of BVOCs from vegetation were calculated with ERA5 meteorology and static land cover using the Model of Emissions of





Gases and Aerosols from Nature (MEGANv2.10, Guenther et al., 2012) on a 0.25°×0.25° grid as monthly mean values as well as monthly averaged daily profiles (Sindelarova et al., 2022). These emissions include 25 BVOC species and chemical

groups. Along with BVOCs from vegetation, SPP also process oceanic biogenic emissions using CAMS-GLOB-OCE dataset, and in this study emissions of dimethyl sulphide (DMS) with a grid resolution of 0.5°×0.5° and hourly temporal resolution were used.

Inputs of size-specific particulate number (PN) emissions are obtained from GAINS model (Amann et al., 2011), which has size-specific particulate number parametrization and calculating particle number emissions (PNE, Paasonen et al., 2016).

PNE and particle size distributions (PSD) used in this dataset are partly based on the European particle number emission inventory developed by Netherlands Organisation for Applied Scientific Research (van der Gon et al., 2009) during the EUCAARI project (Kulmala et al., 2011), but updated, e.g., in terms of road transport emission based on TRANSPHORM database (Vouitsis et al., 2013) and in terms of residential combution based on wide literature review (Paasonen et al., 2016) and references therein). The diameter ranges of the size classes applied in the GAINS emissions are 3–10nm, 10–20nm, 20–

30nm, 30–50nm, 50–70nm, 70–100nm, 100–200nm, 200–400nm, and 400–1000nm. Since the GAINS size distribution is much coarser than that in SOSAA, using the emissions directly in SOSAA would have led to strong artefacts in the final modelled particle size distribution, and to overcome this, a smoothing function was applied to the calculated emissions that preserved the total $PM_1$, $PN_1$ and the bin number concentration as well as possible ($PN_1$ and $PM_1$ error in the smoothed emissions was typically less than 0.5%). The number emissions of the coarse mode (starting from around 230 nm) were gradually

reduced in the attempt to have a better closure between the emitted masses calculated from $PN_1$ and $PM_1$ from ECLIPSE V6b Baseline scenario (Klimont et al., 2017). The scaling is further discussed in the Supplementary material.

Even though CAMS and GAINS datasets were used in this study, SPP can be easily configured to use any type of emissions inventories. For our purposes, the chosen datasets were most suitable as they are readily available and consistent given our modelling domain and general spatio-temporal resolution.

The methods used in SPP to calculate the mean emissions along the trajectory is discussed in more detail the Supplementary material.

### 2.1.3 SOSAA trajectory model

The chemistry and aerosol model is based on the SOSAA (model to Simulate the concentrations of Organic vapours, Sulphuric Acid and Aerosols, e.g. Zhou et al. (2017); Chen et al. (2021); Boy et al. (2022) and ARCA-box (Atmospherically

Relevant Chemistry and Aerosol model, Clusius et al., 2022) models. The model simulates the chemistry and aerosol processes in a column from the surface up to 2.5 km altitude, divided into 45 layers with increasing depth. SOSAA models the vertical turbulent diffusion with the simple Grisogono K(z) scheme, originally developed for the EMEP air pollution model ( Jeričević et al., 2010). The necessary input for it, friction velocity (against which K is scaling linearly) and boundary layer height (BLH) are taken from ERA5 and FLEXPART output, respectively: friction velocity is calculated from northward and

eastward surface stresses, whereas BLH is the average mixing height at the dispersed FLEXPART tracer plume. With BLH




= 1500 m and friction velocity = 1 m⁻¹ s⁻¹ the scheme gives a maximum $K_h$ of 74 m⁻² s⁻¹, occurring at around 315 m, or 0.21×BLH, whereas a minimum was set with k≥0.01 m² s⁻¹, in practice applied above the boundary layer.

Trajectory SOSAA uses a simplified dry deposition scheme where the losses of the aerosols in the first 20 m were calculated using the low and high vegetation leaf area index from ERA5 as scaling factor. Particles were deposited with gravitational

settling, impaction, interception and Brownian deposition by treating the canopy as needle-leaf tree everywhere. The evaluation with SOSAA dry deposition velocities against literature are in the Supplementary material.

Modelling dry deposition of gaseous compounds requires detailed information on the land use types and surfaces. To simplify the modelling setup the dry deposition was approximated with a single loss rate of $10^{-4}$ s⁻¹ in the first 20 m (nominal canopy height) for all gaseous compounds.

Wet deposition, or cloud scavenging was not modelled in this study. As $SO_2$ has a large sink in the cloud droplets, and is subsequently removed with rain-out, omitting wet deposition would lead to overestimation of $SO_2$. At its current state SOSAA-FP does not model aqueous phase particle chemistry and the uptake of inorganic compounds, and to counterbalance this omission in SOSAA-FP, we applied a constant factor of 0.5 for all $SO_2$ emissions.

### 2.1.3.1 Chemistry scheme

The chemistry scheme used in SOSAA for this study is based on a subset of the Master Chemical Mechanism (MCM 3.3.1 Jenkin et al., 1997; Saunders et al., 2003; Jenkin et al., 2015), and is augmented with the Peroxy Radical Autoxidation Mechanism (PRAM, Roldin et al., 2019), which simulates the autoxidation reactions of monoterpenes. The MCM subset was selected so that a reasonable mapping of the CAMS emissions to the chemistry system could be achieved. The selected precursors and their mappings are shown in Table S1.

SOSAA chemistry module solves the time-evolution of the 3542 chemical species and 11334 reactions in a 60 second time step, and is based on the Kinetic PreProcessor (https://kpp.readthedocs.io/en/latest, last access 5 May, 2024, Lin et al., 2023). As the chemistry module is identical to that of the ARCA box, the interested reader is referred to Clusius et al. (2022), which describes the kinetic chemical system, solvers and calculation of photochemical reaction rates in detail.

### 2.1.3.2 Aerosol module

The aerosol module follows that of ARCA box, which uses a sectional particle size distribution and simulates formation of stable atmospheric clusters via the ACDC module (Atmospheric Cluster Dynamics Code, Olenius et al., 2013), condensational growth and Brownian coagulation as described in Clusius et al. (2022). In this study we used 60 logarithmically spaced particle size bins ranging from 1.07 nm to 2 μm in diameter and using the fully stationary method of distributing particles after growth and coagulation. The condensation of 600 lowest volatility organic compounds from the chemistry was

simulated using the APC scheme (Jacobson, 1997; Jacobson, 2002). There are different parametrisations for estimating the volatilities and their temperature dependencies, and in this study, we used the VBS method as suggested in Stolzenburg et al. (2022) (Eqs. 10, 11 and 12). The necessary parameters for the reference saturation concentration at 300 K were those in



Stolzenburg et al. (2018) for products originating from monoterpene peroxy radical autoxidation chemistry, and those in Donahue et al. (2011) for all other compounds. Condensation of sulfuric acid is calculated assuming a zero saturation vapour pressure. The SOSAA aerosol module does not calculate the ion balance and the uptake of water and soluble inorganics, but a rough estimation of the uptake of ammonia and nitric acid to particles was made in order to not underestimate the growth of nucleation and Aitken modes. The uptake of nitric acid was estimated by using the collision rate scaled with relative humidity so that the effective collision rate was

$$CR_{HNO3,eff} = MAX\left(0, \frac{RH-60}{40}\right) CR_{HNO3} \tag{1}$$

The uptake of ammonia was estimated by adding two $NH_3$ molecules for each condensed $H_2SO_4$ molecule and one for each $HNO_3$ molecule when the gaseous $NH_3$ concentrations allowed this. The size-resolved sea salt particle fluxes were calculated following Ovadnevaite et al. (2014), which is a parametrisation that takes into account the sea state with means of the Reynolds number. For this study the parametrisation was simplified so that the Reynolds number (Re) was replaced with wind at 10 m height (U10) by equating the reported mean fluxes as a function of Re and U10 and solving for Re. This simplification does not take into account that the salt particle fluxes are different when the wind is increasing and the sea state is still developing, and when the sea state is matured – whereas the Reynolds number captures this behaviour Ovadnevaite et al., 2014). The sea salt emissions were multiplied by the sea fraction of the land-sea mask. Over the Baltic Sea, which is brackish water and has smaller fetch than the oceans, the sea salt emissions were reduced to 5% of the calculated fluxes.

New particles from the nucleation module, primary particle or sea salt emissions were added to their respective size bins and assigned a composition. As the exact composition of primary particles is unknown, they are assigned a tracer composition, which enabled us to track them and their emission day in the model through all the processes.

The formation of stable clusters through neutral and ion-mediated pathways was calculated by the ACDC module using the $H_2SO_4$–$NH_3$ chemistry (Besel et al., 2020), where the cluster stabilities are based on energies calculated with DLPNO-CCSD(T)/aug-cc-pVTZ//ωB97X-D/6-31++G** level of theory. This dataset can be considered as the best estimate of pure $H_2SO_4$–$NH_3$ clustering efficiency, but it could underestimate the clustering in atmospheric conditions where other components could be taking part in the cluster formation. Other datasets, notably those calculated with the B3LYP/CBSB7//RICC2/ aug-cc-pV(T+d)Z have actually shown better agreement with laboratory experiments (Almeida et al., 2013; Kürten et al., 2016), but as the modelled [$H_2SO_4$] showed some overestimation, the more conservative DLPNO chemistry was used.

## 2.2 Model setup and analysis

In this work, the FLEXPART trajectories were calculated (and the SOSAA model was run) seven days backward. The gas concentrations started from zero with the exception of $SO_2$, CO and $O_3$, which were initialized using CAMS reanalysis data. The particle size distribution is initialized with two modes (with count median diameters at 14 and 52 nm) $PN_{2\mu m}$ 1250 cm$^{-3}$, and therefore we considered the first three days to be spin-up time, and the results shown here are generally focused on the last 96 hours before the station. We simulated the base scenario, where the model input data was used in their nominal values



(with the exceptions discussed earlier). We also tested, how sensitive the modelled CCN concentrations are to a relatively small perturbation in key input emissions and processes. The parameters that were perturbed with a constant factor were a) all biogenic emissions (CAMS-BIO), b) anthropogenic particle number emissions (PNE) and c) the nucleation rates calculated by ACDC. Additionally, we simulated two extreme cases where d) nucleation and e) primary particle emissions were turned off. Where the first three simulations can be thought of testing the model base state, the latter two shift the model substantially from the base state. These simulations are not representing realistic future scenarios, but estimate the effects of anthropogenic particle emissions and atmospheric clustering on CCN concentrations separately, and, when compared with BASE, their effect to each other. The scenarios are summarized in Table 2.

*Table 2: Description and names of the SOSAA simulations used in this study.*

| Scenario name | Description | Motivation |
|---|---|---|
| BASE | Nominal emissions, except $SO_2 \times 0.5$ | Reference case |
| SensiNUC | BASE, but formation rates $\times 3.0$ | Test CCN response to atmospheric cluster formation rates |
| SensiPNE | BASE, but anthropogenic particle emissions $\times 1.20$ | Test CCN response to anthropogenic particle emissions |
| SensiBIO | BASE, but all emissions from CAMS BIO $\times 2.0$ | Test CCN response to biogenic emissions |
| ZeroNUC | BASE, but formation rates $\rightarrow 0$ | Contribution of atmospheric clustering and primary anthropogenic emissions to CCN |
| ZeroPNE | BASE, but anthropogenic particle emissions $\rightarrow 0$ | |

### 2.2.1 Calculation of modelled CCN concentration

The modelled CCN concentrations presented here are calculated for maximum supersaturations between 0.1% and 1.2%, using the κ-Köhler theory (Petters and Kreidenweis, 2007), which relates the water vapour saturation to particle dry and wet diameters taking into account their hygroscopicity, expressed with the *κ*-coefficient. For a given dry particle diameter and κ, the critical wet diameter and the corresponding saturation ratio *S* can be determined from the (local) maximum of κ-Köhler equation:

$$S(D_{p,w}) = \left(1 + \frac{\kappa_{D_{p,d}} D_{p,d}^3}{D_{p,w}^3 - D_{p,d}^3}\right)^{-1} \exp\left(\frac{4\sigma v_w}{R_g T D_{p,w}}\right) \tag{2}$$

where $D_{p,d}$ and $D_{p,w}$ are the dry and wet diameter of the particle, $\sigma$ is the solute surface tension (here value of pure water was used), $R_g$ is the ideal gas constant and $T$ is the droplet temperature (we used the model temperature at ground level to compare with measured CCN).

The particle hygroscopicity κ depends on the composition, and here we used bulk values for the main aerosol components. This simplification can be justified by the unknown activity coefficients of 600 condensing organic compounds, the resulting composition mixture and the generally unknown composition of the anthropogenic particle emissions. To get an idea of the range of uncertainty in the CCN concentration related to the hygroscopicity, a combination of lower and upper κ ranges were used (the chosen κ and the ranges are shown in Table S2).

The κ values for any given particle size in bin *i* was calculated using $\kappa_i = \sum_{j=1}^{N_{comp}} \kappa_j v_{j,i}$, where $v_{j,i}$ are the volume fractions of the components in bin *i*. Using these κ in Eq. (2), the critical maximum supersaturation as a function of particle size and κ




for the model particle size distribution can be calculated. Finally, the $CCN_S$ number concentration for a given supersaturation

$S$ (expressed in %) is the sum of particles whose $S_{crit} \leq S$.

### 2.2.2 Calculation of model response

The focus of this study was to create a model which is able to simulate the CCN concentrations, and then investigate how the CCN may be formed. As the framework consists of many steps which contain averaging over large areas or times, it is unfortunately expected that the model misses any such variability which is beyond the resolution of the input data. Any particu-

lar year and location has special conditions, which are not perfectly captured by the often very generic emission profiles in the datasets. Even with these shortcomings, given the model is in a reasonable proximity with observations, we can test how the modelled CCN responds to different processes and input parameters. A perturbation test gives an estimation of the significance of a single input in a multivariate model. Furthermore, as it turns out, the extrapolations made with the responses are passable estimates of the effects of larger perturbations. For example, everything else being equal, if biogenic emissions in-

crease by a factor of 2, with what factor does the CCN change?

We define the response $R$ as a linear sensitivity to the input. We can write the response $R$ in terms of two model run output $Q_{BASE}$ and $Q_{PERT}$, and their respective inputs $I_{BASE}$ and $I_{PERT}$, obtained from perturbed (PERT) and unperturbed (BASE) model runs, respectively

$$R = \frac{\frac{Q_{PERT} - Q_{BASE}}{Q_{BASE}}}{\frac{I_{PERT} - I_{BASE}}{I_{BASE}}} \tag{3}$$

which simplifies to

$$R = \frac{f_Q - 1}{f_I - 1} \tag{4}$$

where $f_Q$ and $f_I$ are the fractions $Q_{PERT}/Q_{BASE}$ and $I_{PERT}/I_{BASE}$, respectively. In some sense the response describes the model's current state with respect to $I$, including its history, all other input, concentrations, weather conditions, geographic location and the emissions that follow from it. Like the model state, $R$ varies in time and place, and due to the non-linear nature of the model, we assume that it is reasonable to use the linear sensitivity when $f_I$ and $f_Q$ are in proximity of 1 (however, if $f_I = 1$, $R$ is not defined), in other words, we assume this state holds if the model is not perturbed too much. Note that the response is

not necessarily bound between -1 and 1. In the results sensitivity is expressed in percent ($R_\% = R \times 100\%$).

In addition to the linear response, we also tested a power function, where $R_{POW} = \ln(f_Q)/\ln(f_I)$. This function was better suited to predict the effect of the cluster formation rates in the ZeroNUC case, and is further discussed in the Supplementary material.



### 2.2.3 Statistical descriptors used for model evaluation

We use normalized mean bias factor ($B_{NMBF}$, Eq. S3, Yu et al., 2006) and squared Pearson correlation coefficient ($r^2$) to evaluate model performance. In addition, mean fractional bias (MFB, Eq. S4) and mean fractional error (MFE, Eq. S5) were calculated for Figure S2 as recommended by EPA (2007). To test if two distributions have unequal means or medians, we use Welch's t-test and Mood's median test, respectively. Of these, $B_{NMBF}$ is the most useful, as it is statistically robust, symmetric around zero ($B_{NMBF} = 0$: no bias) and readily interpretable: the factor of under- or over-estimation is $f = \left(1 + |B_{NMBF}|\right)^{sgn(B_{NMBF})}$

e.g. $B_{NMBF} = -0.5$ means the model has a bias (of underestimating) by a factor of $(1+0.5)^{-1} = 1 \div 1.5$, whereas $B_{NMBF} = 0.5$ means bias (by overestimating) by a factor of 1.5.

### 2.3 Measurement data for model evaluation

We used data from the SMEAR II (Station for Measuring Ecosystem–Atmosphere Relations; Haataja and Vesala, 1997; Hari and Kulmala, 2005) measurement station, (https://smear.avaa.csc.fi/, last access: Nov 5, 2024, Junninen et al., 2009) to eval-

uate the model results at the end of the trajectories. The SMEAR II station features continuous atmospheric measurements for a boreal forest stand since 1995, and is extensively described in literature (Ilvesniemi et al., 2009; Keronen, 2017; Kolari et al., 2022), and houses a broad range of basic aerosol, trace gas, meteorological, ecological and soil measurement instruments on permanent basis. Beside the observations by the permanently stationed instruments, the station also often hosts intensive measurement campaigns with specialised instrumentation (e.g. Kulmala et al., 2009; Kulmala et al., 2011; Williams

et al., 2011; Petäjä et al., 2016). The station represents boreal zone atmospheric "background" observations, with pronounced local and regional influence from forested areas (Williams et al., 2011), with very few local anthropogenic emission sources but some regional ones (Keronen, 2017). However, long-range transported air masses from continental Europe, British Isles, Eastern Europe, Southwest Russia (including St. Petersburg area), Kola Peninsula and Baltic regions regularly affect the aerosol, VOC and trace gas observations on site (Kulmala et al., 2000; Riuttanen et al., 2013; Patokoski et al., 2015; Keronen,

2017).

The modelling period was selected to be March to October 2018. The choice of the time was largely based on good data availability both for model input and model evaluation, but at the same time there were interesting meteorological features especially with regards to temperature. Compared to 30-year mean, March 2018 was colder and April slightly warmer, while during summer Finland experienced heat waves with lower than average precipitation in May and July, June temperatures

being comparable to 30-year mean. Autumn was warmer than on average.

For this study a subset of comparable observation data was selected as follows. Atmospheric inorganic trace gas measurements from gas analysers (CO, $O_3$, $NO_X$, $SO_2$, Keronen, 2017), the volatile organic compounds (VOC) concentrations continuously measured with a proton transfer reaction mass spectrometer (PTR-MS, Lindinger and Jordan, 1998; Taipale et al., 2008; Blake et al., 2009). Aerosol particle number size distributions between 3–1000 nm diameter observed with with a dif-

ferential mobility particle sizer (DMPS, Aalto et al., 2001) and CCN concentrations with cloud condensation nuclei counter





(CCNC, Roberts and Nenes, 2005; Paramonov et al., 2015). Aerosol particle chemical composition was obtained from using the Aerosol Chemical Speciation Monitor (ACSM, Ng et al., 2011; Heikkinen et al., 2020), which offers a standard speciation into organics, sulfates, nitrates, ammonia and chlorides. When the measurement resolution allowed it, the data was averaged to hourly means before comparison, otherwise daily medians were used. Gas concentration data was averaged from

samples between 4.2 and 125 meters, particle measurements are from at approximately 4 meters height. We emphasize that these data were not used as input in the model, instead the model relies completely on gridded, global emission, concentration and meteorological reanalysis data.

## 3 Results and discussion

### 3.1.1 Comparison of modelled and measured gas concentrations

The modelled and measured gas concentrations were aggregated to daily median gas concentrations at SMEAR II and are shown in Figure 2, while the 8 months and summer medians are shown in Table 3. Before aggregation the modelled hourly values were averaged between 8–50 metres. The best model performance in terms of correlation during the 8 month period is seen in ($r^2$ and $B_{NMBF}$ given in brackets) [acetone] (0.73, –1.95), [methanol] (0.67,–0.06), [monoterpenes] (0.59, –0.76) and [HOM] (0.53, 0.59), whereas smallest $B_{NMBF}$ is seen in [methanol] and [$O_3$], with [HOM] and [monoterpenes] as distant 3rd

and 4th. Worst performers in terms of correlation of daily medians were $H_2SO_4$ (0.01, 2.33), CO (0.12, –4.21), $O_3$ (0.10, –0.09, even though ozone is directly read in from CAMS) and $SO_2$ (0.08, 2.10). Isoprene (0.52, –3.18) concentrations are underestimated quite substantially outside the growth season, whereas the discrepancy is smaller during summertime. The model overshoots $SO_2$ and consequently $H_2SO_4$, even when the $SO_2$ emissions were reduced by half. Ammonia is not routinely measured at SMEAR II, and the modelled $NH_3$ is shown against weekly filter measurements available at EBAS

database (Tørseth et al., 2012), which the model overshoots by a magnitude. As dry deposition is the most significant sink for $NH_3$, followed by in-cloud scavenging and oxidation being negligible (Renard et al., 2004), the simplistic deposition scheme used here could result in accumulation along the trajectory. This means that the cluster formation in the ACDC could be enhanced towards the end of the trajectory by approximately a factor of 3 when compared to measured $NH_3$ concentrations (in STP, $10^7$ cm$^{-3}$ $H_2SO_4$ and 0.1–10 ppb $NH_3$ the formation rates vary between 1–4 cm$^{-3}$ s$^{-1}$) Otherwise this overshoot-

ing has little effect in the model since the gas phase chemistry scheme does not have reactions involving $NH_3$, and the particle phase uptake of it in the model is limited by $H_2SO_4$ and $HNO_3$. Furthermore, the increased cluster formation (which is also due to likely overestimation of $H_2SO_4$) counterbalances the very conservative estimation of the DLPNO-based cluster chemistry.

Modelled daytime median OH concentrations were on average $10^6$ molecules cm$^{-3}$ (the 95 percentile of daily concentrations were typically at $2–3 \times 10^6$ molecules cm$^{-3}$). Long-term OH measurements are not available for the time period, but the result can be compared with earlier SOSAA-modelled concentrations at SMEAR II (Chen et al., 2021), where the modelled daily median concentrations during the time period ranged between $1 \times 10^6 – 6 \times 10^6$ molecules cm$^{-3}$. However, in that study the



modelled OH concentrations were compared with measured concentrations during two earlier campaigns (EUCAARI, from late Apr–May 2007, Kulmala et al., 2011; Williams et al., 2011), and the model was found to overestimate the measured

daily median concentrations, which were consistently below $10^6$ molec cm$^{-3}$. The authors discuss that the measurement height could affect the biases (the measurements were done close to the ground, typically at 4 m height, inside forest canopy). In our work the mixing inside and outside the canopy was not modelled in the same detail, and the difference between the first 10 m and the mean value between 10–80 m is negligible.

Similarly to OH, $NO_3$ measurements were unavailable, but our results, which are mostly under $10^7$ molecules cm$^{-3}$, are in

line with the previous modelling study from SMEAR II, which showed a yearly range of roughly $10^5$–$10^7$ molecules cm$^{-3}$ in 2018 (Chen et al., 2021).

At SMEAR II the HOM are primarily products of mono- (Ehn et al., 2014; Bianchi et al., 2019) and sesquiterpenes (Li et al., 2021; Dada et al., 2023). The lifetime of both terpenes and their oxidation products are short compared to the timescales of long-distance transport. Therefore, the good agreement with the modelled and measured HOM indicates that the emissions

close to the station and the relevant chemistry are consistent with each other. These processes are vital steps in modelling SOA formation, and the results suggest that the model has the potential to model SOA formation properly also upstream along the trajectories, given that the emissions are realistic. The biogenic emissions in CAMS are modelled by MEGAN, but are provided as monthly mean values, together with their mean daily profiles. Therefore, the emissions are missing much of the day-to-day variation, and so will SOSAA. The modelling community would benefit from an open-access dataset with the

gridded global modelled biogenic emissions from MEGAN (or any emission model) in high temporal resolution.




*Table 3: Median concentrations (molecules cm⁻³) of key components during summer (JJA) and total time period (Mar–Oct 2018), and bias ($B_{NMBF}$) between model and measured daily medians. Bias values in bold if FAC2>0.5.*

| | 2018 Mar-Oct median (molec cm⁻³) | | | 2018 Summer median (molec cm⁻³) | | |
|---|---|---|---|---|---|---|
| | SMEAR II | SOSAA-FP | $B_{NMBF}$ | SMEAR II | SOSAA-FP | $B_{NMBF}$ |
| OH day | | $9.92\times10^5$ | | | $1.03\times10^6$ | |
| OH night | | $5.32\times10^4$ | | | $9.45\times10^4$ | |
| O₃ | $8.55\times10^{11}$ | $8.29\times10^{11}$ | **−0.09** | $8.09\times10^{11}$ | $8.85\times10^{11}$ | **0.06** |
| NOₓ | $1.18\times10^{10}$ | $6.40\times10^9$ | −0.97 | $8.28\times10^9$ | $6.37\times10^9$ | **−0.24** |
| NO₃ | | $3.67\times10^6$ | | | $2.70\times10^6$ | |
| NH₃ | | $3.96\times10^{10}$ | | | $3.58\times10^{10}$ | |
| SO₂ | $1.10\times10^9$ | $5.01\times10^9$ | 2.18 | $1.05\times10^9$ | $4.39\times10^9$ | 3.49 |
| H₂SO₄ | $2.86\times10^5$ | $1.24\times10^6$ | 2.32 | $4.81\times10^5$ | $1.36\times10^6$ | 2.39 |
| Methanol | $4.11\times10^{10}$ | $4.57\times10^{10}$ | **−0.06** | $8.13\times10^{10}$ | $8.17\times10^{10}$ | **0.03** |
| Monoterpenes | $4.61\times10^9$ | $3.19\times10^9$ | **−0.77** | $8.93\times10^9$ | $6.06\times10^9$ | **−0.74** |
| HOM | $5.50\times10^7$ | $7.66\times10^7$ | **0.58** | $1.11\times10^8$ | $1.44\times10^8$ | **0.67** |
| Isoprene | $2.12\times10^9$ | $2.75\times10^8$ | −3.18 | $4.62\times10^9$ | $1.12\times10^9$ | -2.86 |
| Benzene | $1.41\times10^9$ | $5.25\times10^8$ | −1.21 | $8.71\times10^8$ | $3.48\times10^8$ | −0.93 |
| Toluene | $2.38\times10^9$ | $2.52\times10^8$ | −8.19 | $3.67\times10^9$ | $2.54\times10^8$ | -14.3 |
| CO | $3.44\times10^{12}$ | $6.32\times10^{11}$ | −4.20 | $2.87\times10^{12}$ | $7.63\times10^{11}$ | -2.59 |





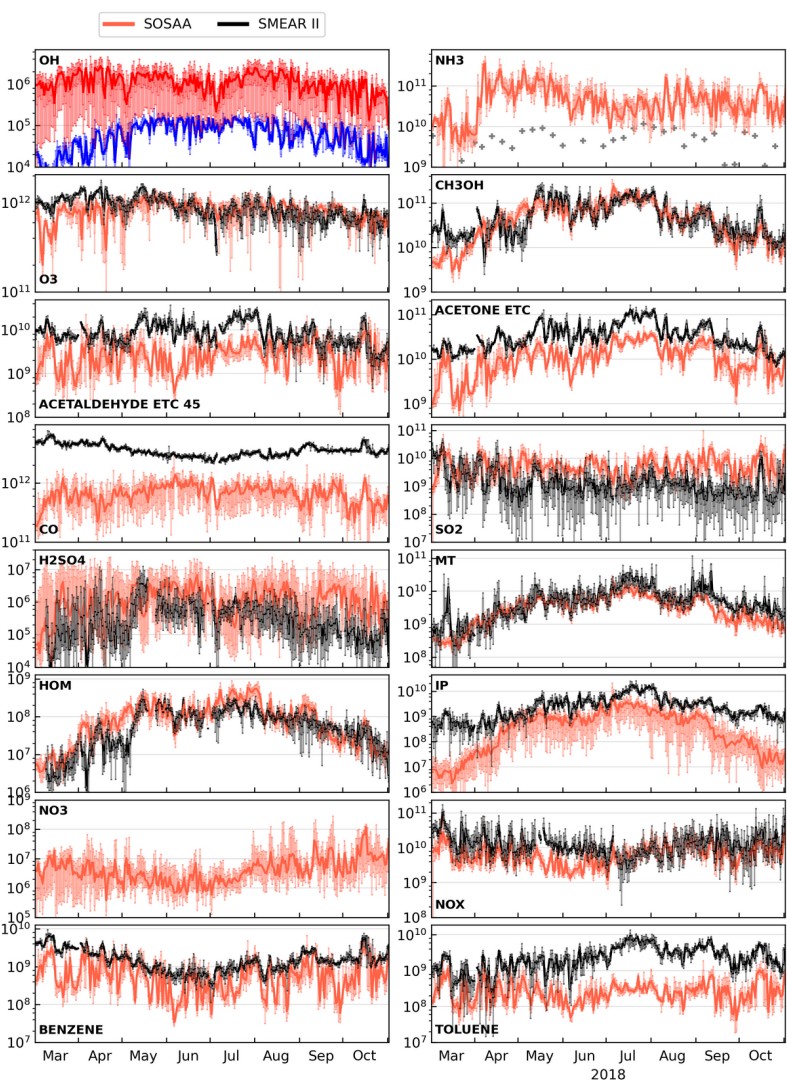

*Figure 2: Gas concentrations at the SMEAR II station in the SOSAA model (red) and available measurements (black). OH concentrations show the daytime in red and night values in blue. The ranges show the daily interquartiles. All concentrations are shown in molec cm⁻³. Ozone concentrations are not modelled by SOSAA, but instead are read directly from CAMS. NH₃ measurements come from EBAS filter measurements with 1 week time resolution.*



### 3.1.2 Aerosol size distributions and composition

Fig. 3 shows the modelled and measured particle size distributions at SMEAR II (panel a) and the ratios between the two (panel b). The fractions were calculated from the size distributions (the measurements were first interpolated to match the modelled bins), so that the fraction $E$ for bin $i$ was defined as $E_i = \frac{N_{i,model}}{N_{i,meas}}$. Figure 3b also shows on the right axis the time-averaged (12 days running mean) fractions for the comparable total size distribution 3–1000 nm and three subclasses; 3–30 nm, 30–280 nm and 280–1000 nm, approximately corresponding to nucleation, accumulation and coarse modes. The total geo-
metric mean of the fractions of modelled and measured number concentrations in total, nucleation, accumulation and coarse modes are 0.89, 1.70, 0.48 and 0.94, respectively, meaning the model generally generated more particles in nucleation mode and less in other modes, with clear exceptions in July and early September, when the observations showed a markedly de-creased nucleation mode which was practically absent in the model. This period has been discussed in earlier studies where the July heat wave resulted in decreased gross primary production in large parts of central and north-western Europe, when
compared with earlier years. Heikkinen et al. (2020) report a high SOA loading in July, but also an increased anthropogenic organic component in $PM_1$. We attribute the weak new particle formation in the model to elevated temperatures and condensation sink due to primary emissions upstream of the trajectories (Fig. S6), which effectively reduces cluster formation in the ACDC module (these are discussed in the supplementary, see also Fig S5 and S6). The increased particle number emissions along the trajectory in mid-July to early August coincide with the decreased nucleation rates in the model.
While the number emissions especially 3 days prior to arriving at station were among the highest in the period, similar emissions were observed in March, but with much less nucleation inhibition. This could to be explained by the lower temperatures in March. Other factors such as $NH_3$ and $SO_2$ emissions along the trajectories did not markedly differ in July from adjacent months (Fig S8).






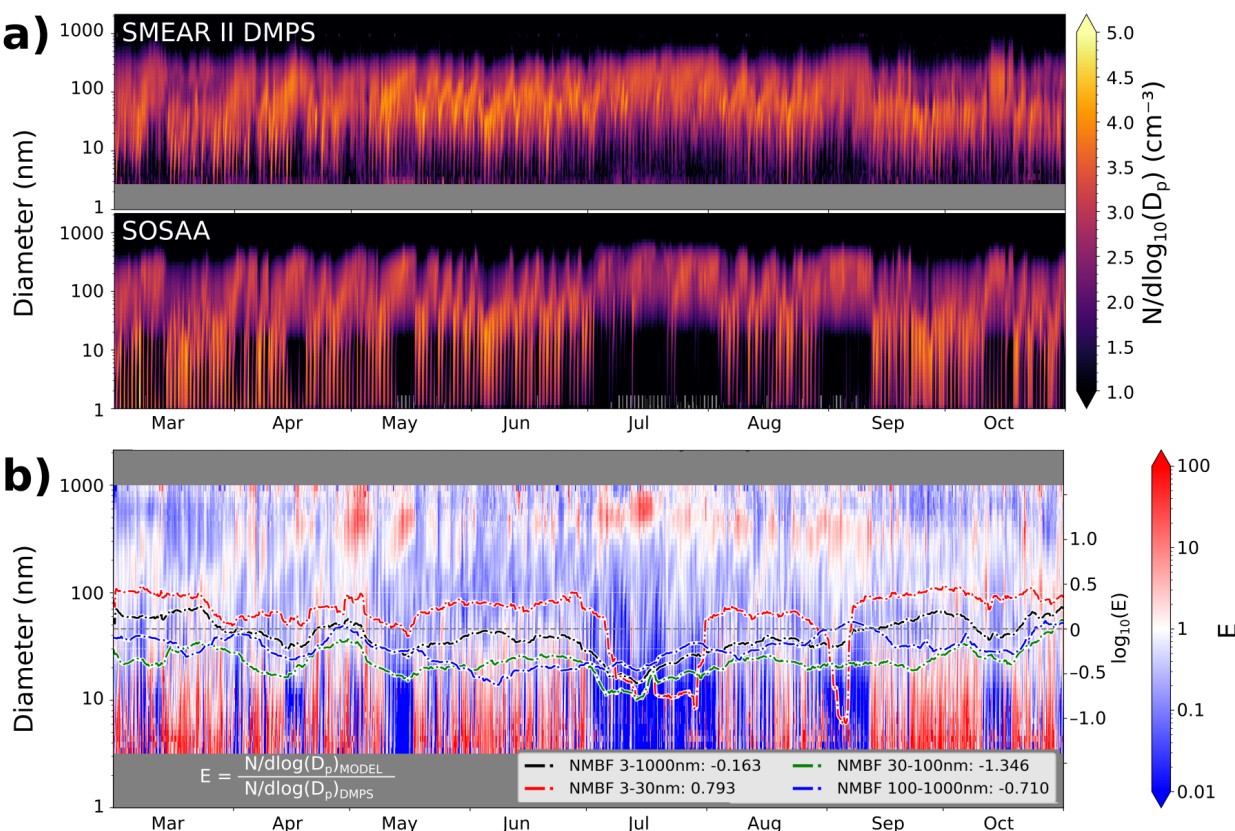

*Figure 3: Modelled and measured (DMPS) particle size distributions from the study period (panel a). Panel b shows the 12 day-running mean of the model error E (Model / Measurement), along with the $B_{NMBF}$. The $B_{NMBF}$ of total comparable size range (3–1000nm) is -0.163 (indicating ca. 14% underestimation), while the model on average overestimates nucleation mode number concentration.*

Figure 4 shows the mean and median (with interquartiles shaded) size distributions from the 8-month period. These figures show the same overshooting in the nucleation and underestimation in the larger modes. While this could be partly explained by the lower sensitivity of the DMPS instrument in the smallest size range, it is possible that the model bias is due to insuffi-

cient growth of the nucleation mode, leading to underestimation of the coarse mode particles. Possible reasons for this could be a significant underestimation of low-volatility organic vapours, missing particle phase reactions which could decrease volatility of organic compounds and thus reduce their later evaporation, missing cloud processing or insufficient partitioning of inorganic compounds. Fig. 4 shows that the perturbation run SensiBIO was the only one where the size distribution shifted towards larger sizes (especially noticeable in summer months, see Fig S9). However, the gas phase concentrations of

HOMs were relatively well captured in the BASE, and the particulate ammonium and nitric acid were already overestimated




when compared with ACMS measurement. This leaves the cloud processes or particle phase reactions as one possible addition to the model which could reduce the difference in the modelled and measured size distributions. Whether either of these would markedly shift the nucleation mode, which is well beyond activation size, and where SVOCs accumulate weakly, is unclear. Cloud droplets are a sink for the particles below activation size (Pierce et al., 2015), and this missing sink could par-

tially explain the overestimation of the nucleation mode, although they might not be deposited in rainout in case of cloud droplet evaporation.

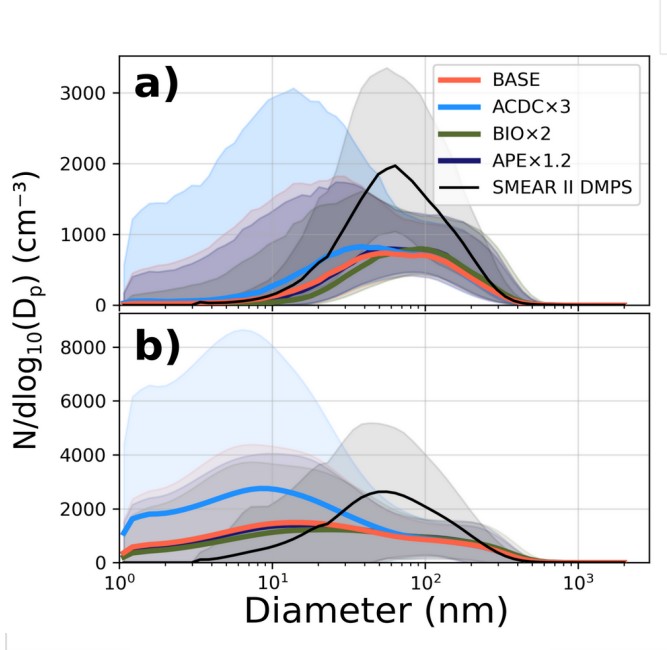

*Figure 4: Median (panel a) and mean (panel b) measured (DMPS) and modelled (BASE and perturbation simulations) particle size distributions at the station.*

We also compared the aerosol mass composition with ACSM measurements from SMEAR II. SOSAA stores detailed information of the secondary organic aerosol, but a substantial fraction of the aerosol loadings in the model comes from an-

thropogenic particle emissions, whose composition varies, and is not known in sufficient detail. Here we assumed that the PNE are primarily composed of organic compounds, black and brown carbon and sulfates. The ACSM shows organic aerosol, sulfates, nitrates, ammonium and chlorides, but cannot measure (refractive) black carbon. For Fig. 5 we attributed the modelled SOA and 85% of the primary particle mass to organic aerosol, all sulfuric acid and the remaining 15% of primary





particle mass to sulfates, nitric acid to nitrates, particle phase ammonia to ammonium and 60% of sea salt mass to chlorides.

The majority of the measured and modelled particles are organics and sulfates. The model is able to capture the time series of the total mass to a good degree; for the 24 h running mean $B_{NMBF}$: 0.17 ($r^2$: 0.53), and by component OA: 0.01 ($r^2$: 0.70), $SO_4$: −0.40 ($r^2$: 0.40), $NO_3$: 2.05 ($r^2$: 0.31), $NH_4$: −0.22 ($r^2$: 0.45), Cl: −0.17 ($r^2$: 0.02). For the sum $OA+SO_4$ the $B_{NMBF}$ was 0.04 ($r^2$: 0.62).

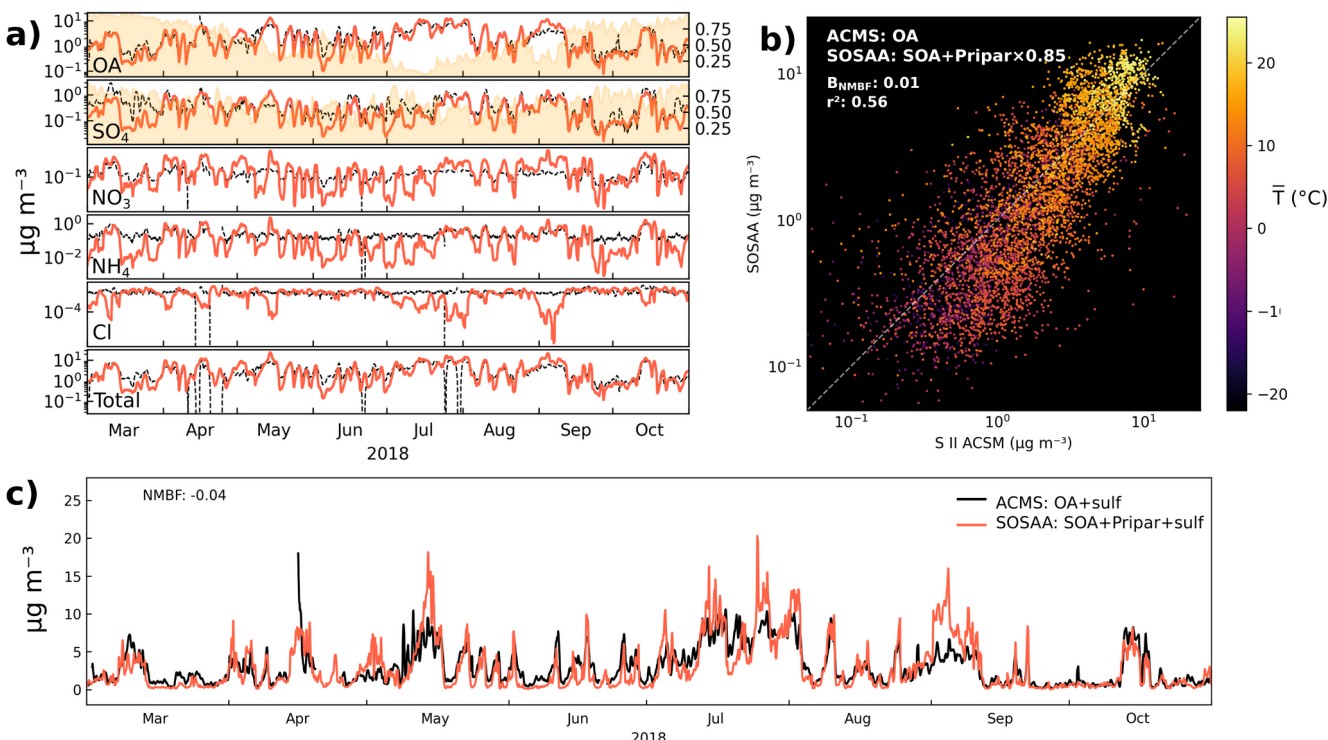

*Figure 5: Panel a: 24-hour running mean of the time series of mass composition from ACMS measurements. For this figure 85% of the primarily emitted particle mass in the model was attributed to organic aerosol, and 15% to sulfates. The yellow shading with the axis on the right shows the modelled mass fraction coming from the primary particles. Panel b: The scatter plot of 3-hourly values organic aerosol of $PM_1$ from ACMS and SOSAA (obtained similarly as in in panel a), coloured with mean temperature during the last 4 days along the mean trajectory. Panel c: The unfiltered 3-hourly ACMS OA+Sulfates and SOSAA primary particles + SOA + sulfates.*

Figure 6 shows the size-resolved relative fractions of the particle composition in the SOSAA model. The secondary aerosol mass is the dominant fraction in the 1−10 nm range (around 80% of mass) and comprises more than 60% of the mass in the



10–100 nm diameter range, especially in the summer months when the secondary mass is on average between 65% to 85% of the total mass in the size range. For sizes over 200 nm in diameter, primary emissions start to dominate the mass, with a tail of sea salt from approximately 500 nm diameter upwards. Figure 7 shows the absolute and relative mass fractions of

PM$_1$. Measurements of size resolved aerosol composition from SMEAR II station exist from March–April 2003, where days with cleaner northern air (associated with higher likelihood of new particle formation events) showed high fraction of organic mass from terpene oxidation products in nucleation and Aitken modes, and organics, nitrates and sulfates in the accumulation modes (Allan et al., 2006). However, the limited resolution, short time period, and 15 year time difference in that study makes quantitative comparison with our results impractical.

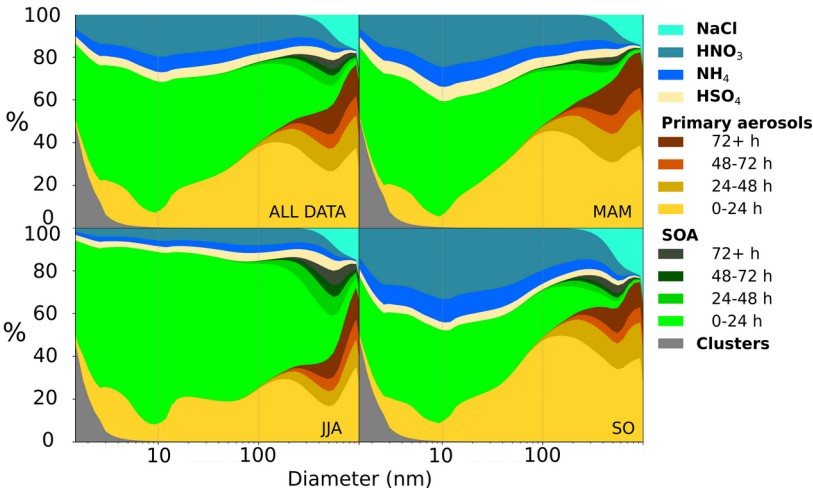

*Figure 6: Size dependent relative mass composition of the modelled particles for the whole period, and spring, summer and autumn months. Primary aerosols and SOA is classified by their age in the final PSD.*






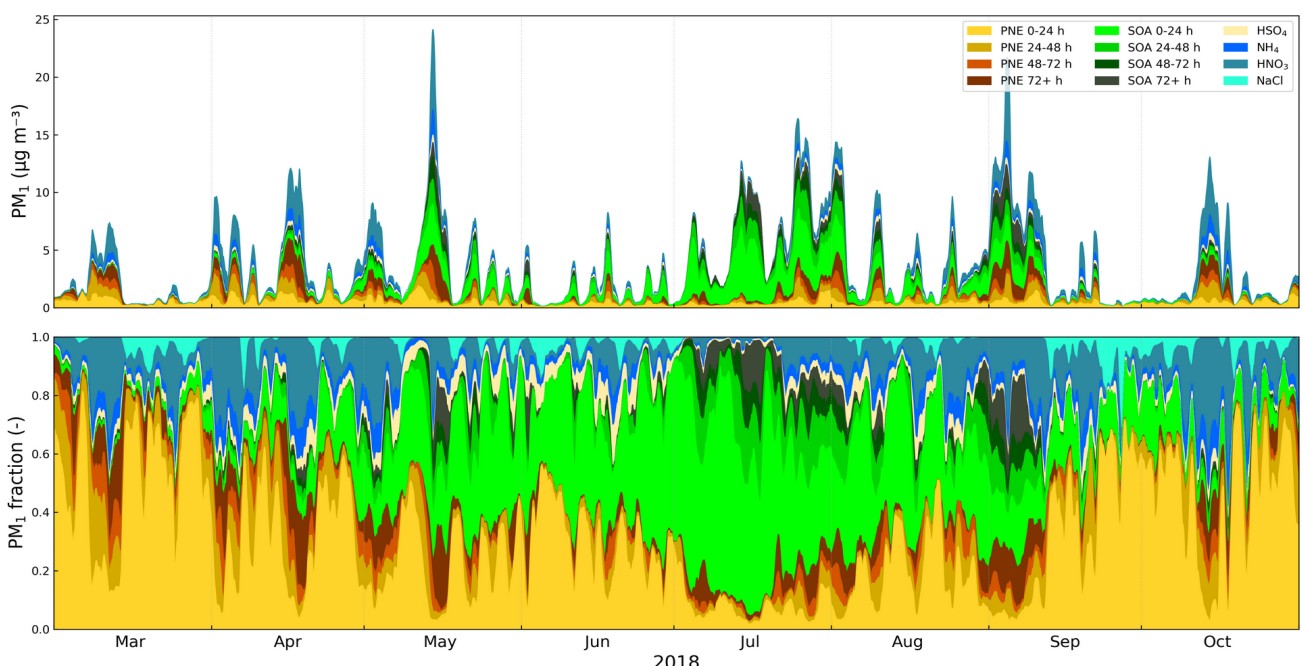

*Figure 7: Main composites of modelled $PM_1$ (top panel) and mass fractions (bottom panel), 24 hour moving average of hourly data.*

### 3.1.3 CCN concentrations

Figure 8 shows the modelled CCN together with CCN measurements at SMEAR II, for three maximum supersaturations. The measured 8-month time series shows large variation, generally higher median concentrations in the summer months and lower in the two autumn months. Summertime (JJA) median of measured $CCN_{0.5\%}$ concentration was approximately +28% of the 8-month median, autumn (SO) approximately –60% of total median concentration, spring (MAM) being close to total median. In the 0.1% supersaturation class only autumn concentrations differed significantly (–40%, p<0.01) from the total median. The model underestimates the concentrations of the 0.2% and 0.5% supersaturation classes, and overestimates the 0.1% class, and does not capture the statistically significant increase in the summertime in the larger supersaturation class but does so with the autumn decrease. The modelled median CCN in the supersaturations above 0.5% showed in practice no change in the summertime (±1 %) and only a weak, statistically insignificant decrease in the autumn. As the measurements for the time period were available only for up to 0.5% supersaturation, this cannot be verified against observations, but it



seems reasonable to assume that the same trends that were observed in the 0.5% supersaturation class would continue even in smaller particles. This means that although the model is, on average able to capture the shorter time scales, as evident from the satisfactory correlation of the daily median in all compared supersaturation classes (r² was 0.51, 0.62 and 0.59 for 0.5%, 0.2% and 0.1% supersaturation CCN, respectively), the seasonal dynamics of CCN remained elusive, and the increase in the summertime smaller diameter CCN is missed. In the 0.1% supersaturation CCN the model has least bias in the summer months, which could be an indication that the coarse mode particles (from primary emissions) are overshooting, and, as they are sinks to smallest particles, this would lead to underestimation of the higher supersaturation classes. However, the time series shows that in the same time periods when the model overestimates CCN 0.1%, the other classes have very little bias (spring) or increase their overestimation (Aug–Sept). In general both modelled and measured CCN correlate with primary emissions – r² between median PNE 0–4 days before the station and CCN at the station was 0.54 for modelled $CCN_{0.1\%}$ and 0.43 for measured $CCN_{0.1\%}$; same comparison between PNE and $CCN_{0.5\%}$ showed r² of 0.30 and 0.22 for modelled and measured CCN, respectively.

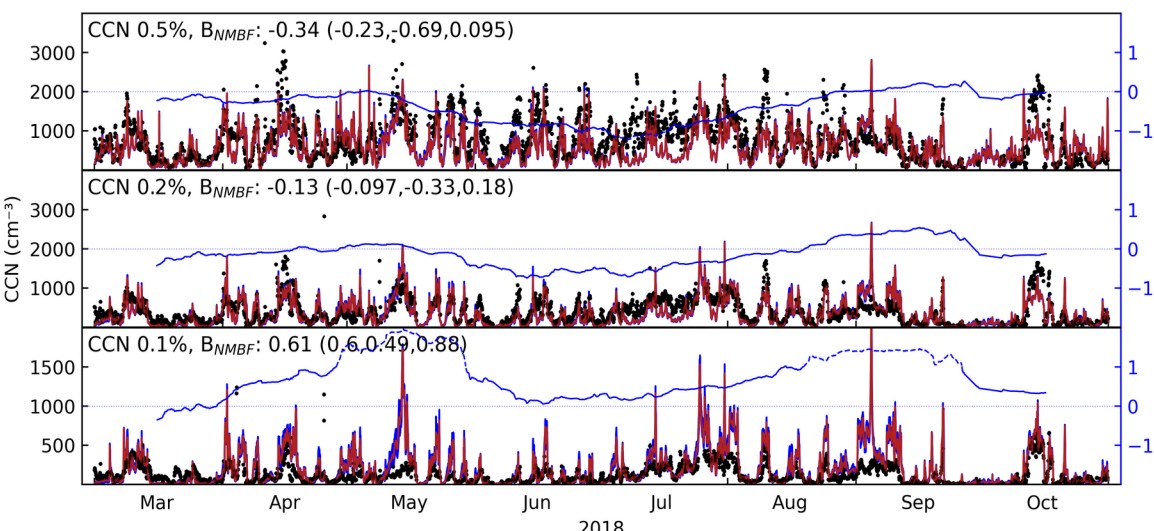

*Figure 8: Comparison of three classes of measured CCN concentrations (black dots) with SOSAA model results (red line); 0.5%, 0.2% and 0.1% supersaturation. In addition to the model line, a blue shading is shown the CCN is calculated using the upper and lower limits of the κ-values of the components (see Table S2), giving an estimate of the uncertainty that is related to the component activities. For the most part the uncertainty due to κ is too small to stand out from the figure. Although the model correlates well with measurements, as shown in the scatter plots comparing the measured and modelled daily median it moderately underestimates the concentrations of the 0.2% and 0.5% supersaturation classes, and markedly overestimates the 0.1% class. The axis on the right shows the running $B_{NMBF}$, with 30-day window, solid line marks periods where model is within a factor of two of measurements.*







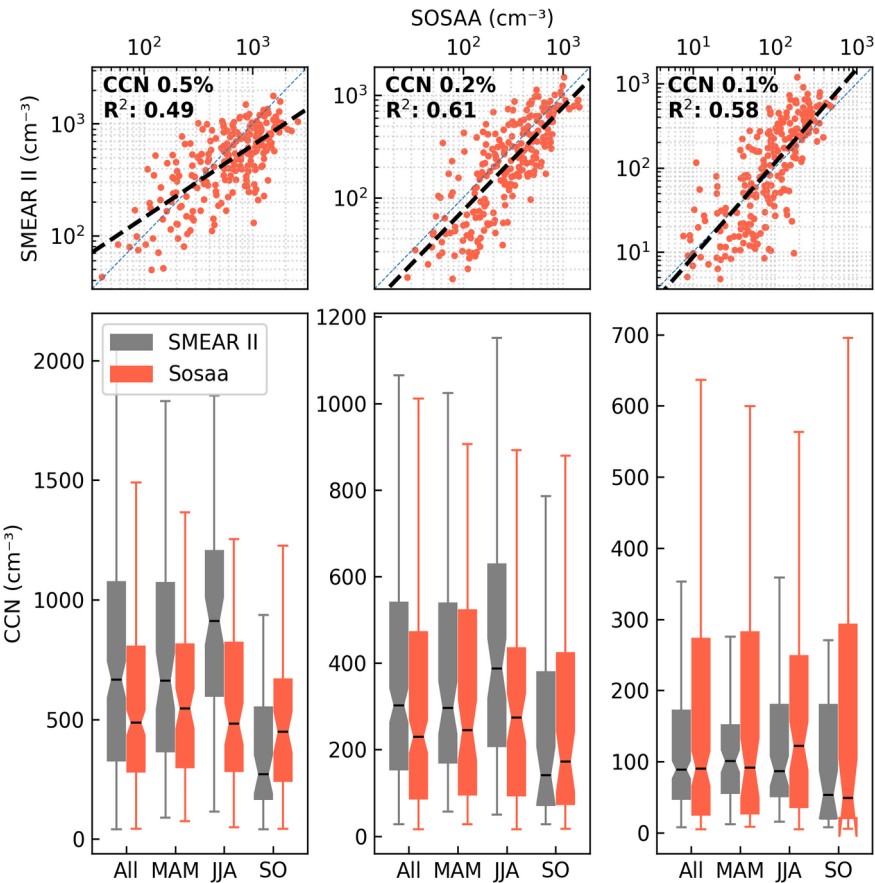

*Figure 9: Scatter (top row) and box plots (bottom row) of modelled and measured daily median CCN number concentrations of three supersaturation classes (left column: $CCN_{0.5\%}$; middle: $CCN_{0.2\%}$; right column: $CCN_{0.1\%}$). Box plots show data for all days and those of spring (MAM), summer (JJA) and the early autumn (SO).*

## 3.2 Model sensitivity: CCN response to selected parameters

The model has shown good potential in capturing the variability of local gas concentrations as well as particle concentrations and composition, including CCN concentrations. The following sensitivity tests investigate the model response to changes in some key parameters. Figure 10 shows the responses of three CCN supersaturation classes (1.0%, 0.4% and 0.2% maximum supersaturation) to the perturbations in the three "Sensi" runs (see Table 2). In all classes, the model CCN are generally most responsive to the primary particle emissions, followed by BVOC emissions, whereas changes in cluster formation rates show the least response. However, the responses show large variation in seasonal and synoptic time scales, which could relate to the seasonal variation of BVOC emissions, as well as the history of the trajectories, namely which geographical areas and



conditions (meteorological, emissions) they originate from. The distributions of the responses, shown on the right panels in Fig 10, show that all parameters have, on average, mostly a positive impact – that is – increasing the variable increases CCN concentrations. However, all parameters occasionally show negative responses, meaning that an increase will decrease CCN concentrations. This is most evident in the case of cluster formation rates, where for the lowest maximum supersaturation

(largest diameter particles), increased nucleation rates show a negative response in CCN concentration, although the effect is much smaller in either direction than those of PNE or BVOC emissions. This is in line with some modelling studies where high concentrations in the nucleation mode affected low volatility vapour concentrations and hindered the growth of larger particle sizes, resulting in less CCN in the lower supersaturation range, typically observed in stratiform clouds ( Roldin et al., 2019; Patoulias et al., 2024). Cluster formation rates have a positive but surprisingly small effect to higher maximum super-

saturation (smaller diameter) CCN. The $R_{NUC}$ stays in general below 10 % (8% for the exponential effect $R_{NUC,POW}$), meaning that to approximately double the CCN concentrations, more than ten-fold increase in the formation rates is needed, whereas a ten-fold decrease would decrease CCN by less than 10%. Figure 11 shows the diurnal profiles of the responses, revealing a more detailed picture where daytime CCN concentrations show considerably higher sensitivity to cluster formation rates, and lower to particle number emissions, when compared with the seasonal medians. Interestingly, changes in BVOC emissions

affect CCN in a similar, if more benign, diurnal pattern to particle emissions. It is not immediately clear how to interpret this behaviour, but during summer months, when emissions peak, the response is strongest in the lowest supersaturation class, which at least implies that the effect of BVOC emission is strongest in the particles that are in the upper range of the Aitken mode, able to accumulate mass via compounds that are low- or semi-volatile, as opposed to extremely low volatility vapours. This would favour the primary particles, which are already in this size range upon emission.



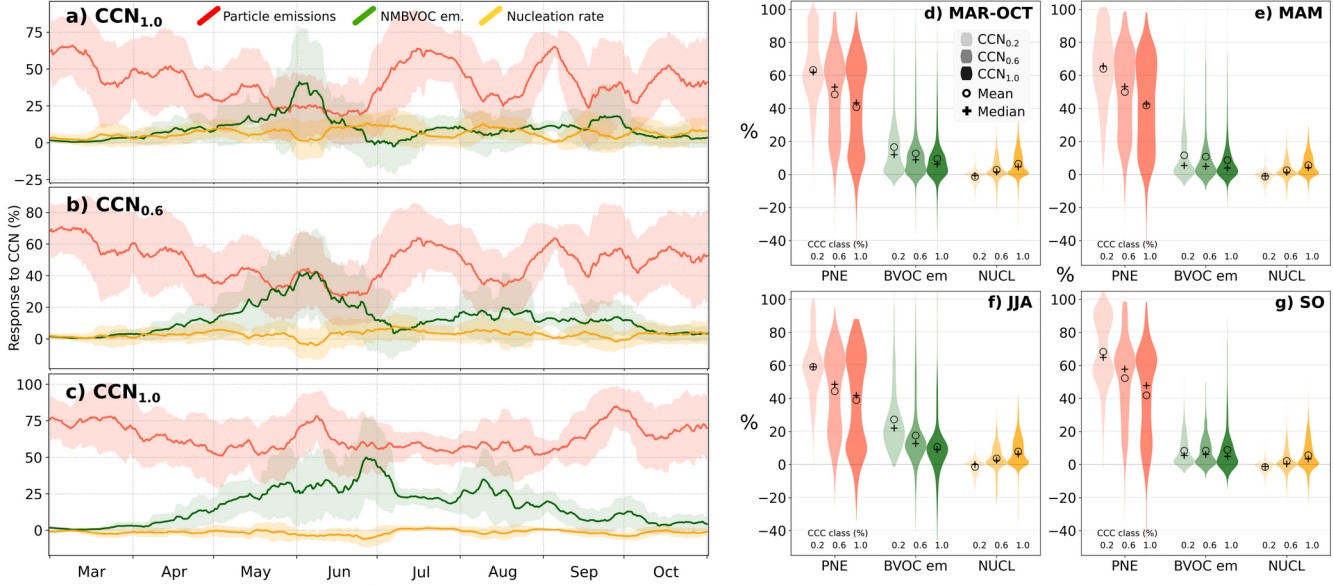

*Figure 10: The response of CCN number concentration from three supersaturation; a) 1.0 %, b) 0.6% c) 0.2 %) to changes in model parameters, 12 days running mean. The three input parameters were varied by multiplication factor f_I 3.0, 1.2 and 2.0 for ACDC J-rate PNE and BVOC emissions, respectively. The shaded areas show the +/− standard deviation for the 12 days window used in the running mean. Panels d–g show the grouped distribution of the responses in a–c as violin plots for the full study period (d), spring (e), summer (f) and autumn (g).*



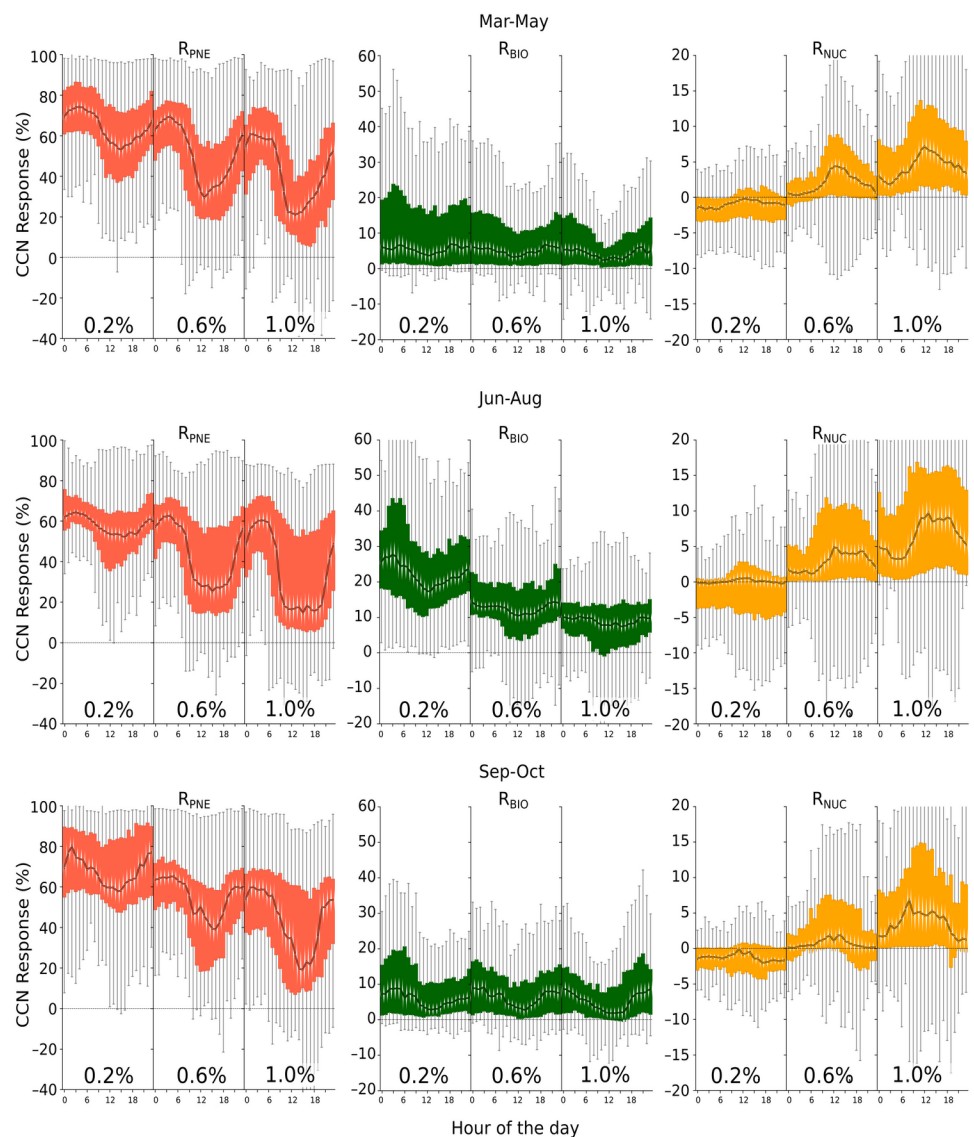

*Figure 11: Diurnal profiles of the CCN responses for 0.2–1.0% supersaturation classes for spring (top row), summer (middle row) and early autumn (bottom row). Colored area shows 25%–75% range of the data, whiskers show 5%–95% range. Note the different scales in the vertical axes of $R_{PNE}$, $R_{BIO}$ and $R_{NUC}$.*

Figure 12 shows the differences in the ZeroPNE and ZeroNUC CCN concentrations, compared to BASE for the total dataset, while Figure 13 shows the monthly distributions along with 12-day running mean. Setting the formation rates to zero de-





creases CCN 1.2% on average by 36%. The same indicators in the 0.1% supersaturation are +29% increase, but due to large variation of the results the differences were (with exception of June) statistically insignificant. The large variation in the effect of cluster formation underpins the complex dynamics of the atmospheric particles and highlights the caveats of expressing the contribution of different processes in a single figure. Removing PNE from the model impacts decreases the coagulation sink, thereby increasing the cluster formation rates (on average by approximately threefold), and these changes in Zero-

NUC are shown in Fig. S4 and further discussed in the Supplementary material.

Removing all particle number emissions from the model had on average decreasing effect in all CCN classes, strongest in the $CCN_{0.1\%}$, where the change was $-77\%$ ($\sigma=21\%$) and smallest in the $CCN_{1.2\%}$ class with a change of $-26\%$ ($\sigma=37\%$).

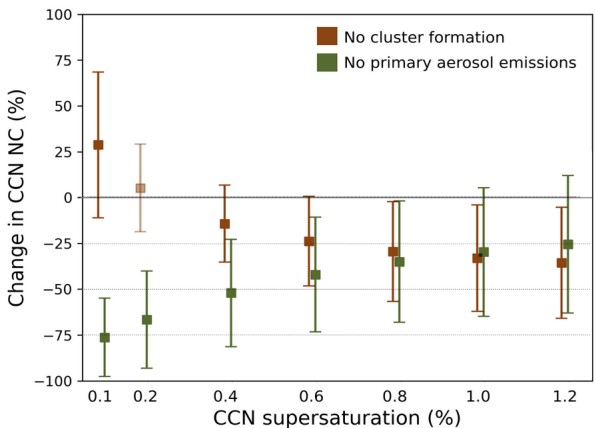

*Figure 12: Changes in CCN number concentrations in Zero-simulations, compared to BASE.*





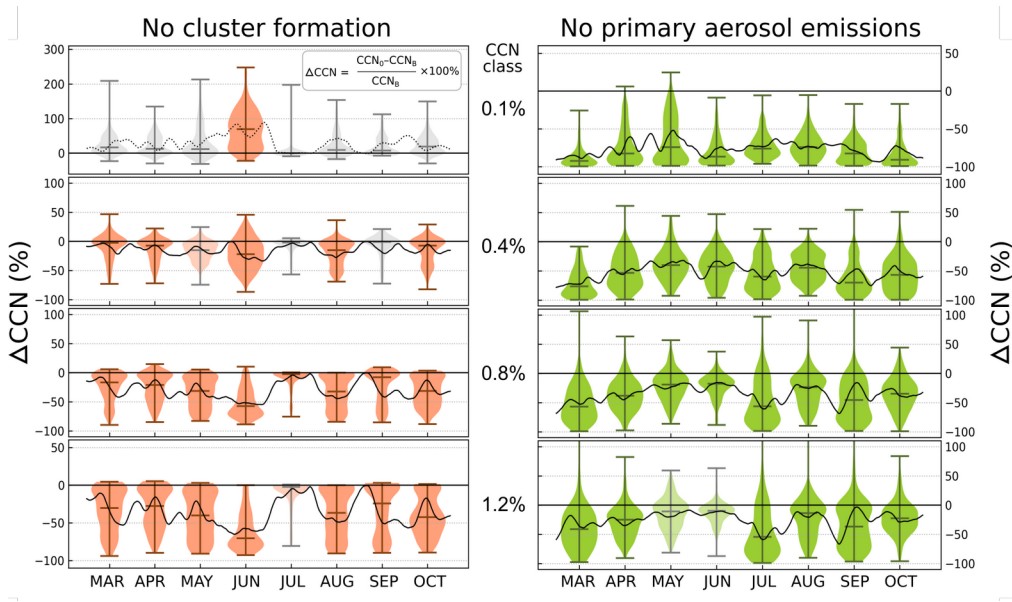

*Figure 13: The relative change in CCN concentrations in two model runs ZeroNUC (left panel) and ZeroPNE (right panel). The 3-hourly modelled CCN concentration at the end of the trajectory (SMEAR II) are grouped in monthly distributions, shown as violin plots. Coloured distributions have p-value<0.001 in the Welch's directional t-test, while light coloured data has p<0.01. For gray distributions p>0.1. The black curve shows the 12-day running mean of the difference between Zero-simulations and BASE. This figure corresponds to data in table 4.*

*Table 4: Average difference in modelled CCN number concentration supersaturation classes in the two Zero-runs, compared with BASE, one where primary anthropogenic particle emissions were turned off, and one where cluster formation (nucleation) was turned off. Welch's one-sided t-test p-value is show, where it was larger than 0.001. The direction of the t-test was chosen by the sign of the mean. The intervals show one standard deviation of the calculated values. This table corresponds to data in figure 13.*





| CCN SS | All months | Spring | Summer | Autumn |
|---|---|---|---|---|
| | | **Primary particle emissions set to zero (ZeroPNE)** | | |
| 0.10% | −77± 21% | −75± 26% | −75±18% | −82±17% |
| 0.40% | −52± 29% | −53± 31% | −48±27% | −59±30% |
| 0.80% | −35± 33% | −36± 32% | −33±33% | −39±35% |
| 1.20% | −26± 37% | −25± 37% | −26±37% | −27±37% |
| | | **Cluster formation set to zero (ZeroNUC)** | | |
| 0.10% | +29±40% p=0.066 | +28±37% p=0.106 | +33±47% p=0.124 | +22±27% p=0.379 |
| 0.40% | −15±21% | −14±20% | −16±22% | −13±21% |
| 0.80% | −30±27% | −29±25% | −31±29% | −28±27% |
| 1.20% | −36±30% | −35±28% | −36±33% | −26±30% |

## 3.3 Origins and history of modelled CCN

Figure 14 shows the geographical origins of the air masses as azimuths and straight line distance from the SMEAR II station for the last four days before arriving at the station, as well as some relevant model variables for CCN formation. For discussion, panel b also shows the modelled [$CCN_{0.2\%}$] and [$CCN_{1.0\%}$]. This overall view on the study period serves as a basis when we try to identify the origins (both geographical and processes) that lead to CCN. In March, cluster formation rates were average throughout the period, with low BVOC emissions. Early March peak in CCN is accompanied with some BVOC and notable PNE and cluster formation, origins from south-east, whereas second half of March saw below average CCN, PNE and BVOC, with average cluster formation, and origins from north-westerly direction. With increasing temperatures in April the BVOC emissions pick up, cluster formation is strong, but the peaks in CCN coincide more with elevated PNE (and BVOC). The turn of April–May shows elevated cluster formation rates 3–4 days prior to station, followed by increasing BVOC emissions and decreasing particle emissions during the last two days, and airmasses originating from west of SMEAR II. The resulting [CCN ] peak in early May can be thought to show a typical pathway from new particle formation to CCN, whereas the peak in mid-May, while still showing strong BVOC emissions, seems to mostly originate from PNE. Otherwise, much of the April–June period includes trajectories with westerly to north-westerly component, formation of clusters and average [$CCN_{1.0\%}$]. Especially in June primary emissions were low, cluster formation along trajectories generally lasted for days with BVOC emissions overlapping, indicating that much of the CCN was likely formed from NPF (as also indicated by the Zero-simulations). In contrast, July and early September mostly saw periods with weak cluster formation, elevated BVOC and primary particle emissions, related to south-easterly origins of the airmasses, coinciding with some of the highest CCN concentrations in the studied time period. The different pathways of CCN origins are also supported by the fractions of [$CC_{N0.2\%}$]/[$CC_{N1.0\%}$], where low fractions are seen in end of March, June mid-August and early October, coinciding with low PNE and suggesting formation of CCN via NPF.





*Figure 14: Diagrams showing the 4-day history of the air masses and some model parameters relevant to CCN formation.* **a)** *azimuth, i.e. bearing from SMEAR II;* **b)** *straight line distance (km) from the station to the trajectory mean, superimposed are the 30-day and 3-day running mean of [CCN$_{0.2\%}$] and [CCN$_{1.0\%}$] at the station (right axis in cm$^{-3}$);* **c)** *temperatures (C°), note that the color scale only covers the range between –10°C and 30°C;* **d)** *formation rates from the ACDC H2SO4×NH3 module;* **e)** *terpene (mono and sesquiterpenes) emissions;* **f)** *total primary anthropogenic particle number emissions;* **g)** *condensation sink of sulfuric acid, used as a scaling factor to estimate coagulation losses of interstitial molecular clusters in the ACDC module.*





Figure 15 shows the geographical origins of the airmasses in the summer (June–August), as the density of the FLEXPART Source-Receptor Relationship values, grouped in sectors where each sector is 1000 km further away from the SMEAR II station. For clarity, the first 200 km radius is left out as the direction of advection at the station is not relevant in this context. The trajectories were divided in two groups, where the top (bottom) row is an aggregation of the trajectories that showed above (below) median CCN 0.4%. For the most part the above median CCN trajectories showed high SOA and primary

particle mass. The below median trajectories show the typical origins that favor NPF events, while the above median trajectories have a southerly component, favouring BVOC but also anthropogenic emissions, although 3–4 days prior to station the airmasses are quite dispersed and also show some NPF favoured North-Westerly component.

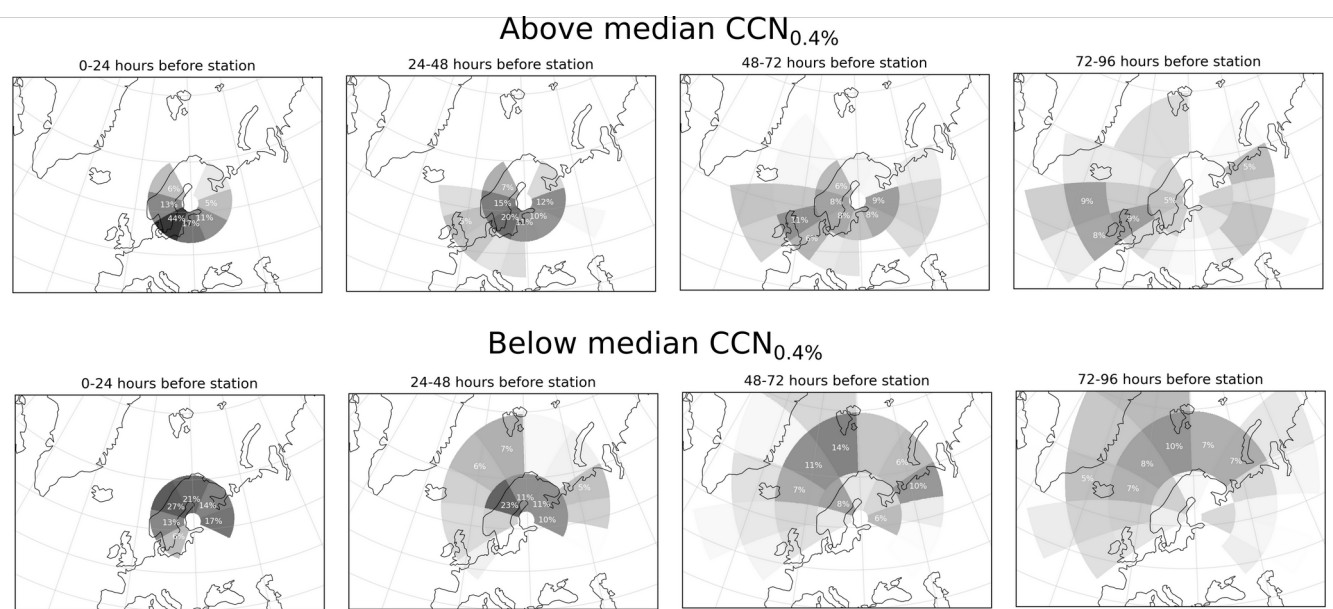

*Figure 15: Origins of airmasses 0–4 days prior to arriving at the SMEAR II, based on the FLEXPART Source-Receptor relationship output. The trajectories for the upper and lower rows were selected by the modelled $[CCN_{0.4\%}]$ so that top row shows trajectories which brought more than JJA median, and bottom row trajectories whose [CCN] was below JJA median.*

In Figure 16 we present scatter plots and correlations ($r^2$) of summertime modelled [CCN] and the discussed model parameters: cluster formation rates, terpene emissions and particle number emissions in three size classes: 3–1000nm, 1–10nm and

10–40 nm. For the cluster formation rates, correlations are also calculated with respect to airmass history, where the final [CCN] is compared against median formation rates 0–1, 1–2, 2–3, 3–4 and 0–4 days before the trajectory arrives at the station (similar view for other parameters, see Figures S12–S13, in the Supplementary material). Here the lack of any correla-





tion between cluster formation rates and CCN concentrations (consistent with the weak CCN responses to cluster formation)
is surprising given the results of the ZeroNUC simulation, but this can be at least partly explained by the diurnal nature of
NPF (Fig. 11). The negative slope of $CCN_{0.2\%}$ and cluster formation rates is partly related to the inhibition of cluster forma-
tion by the coarse mode loadings and the correlation of PNE with $CCN_{0.2\%}$. On the other hand, both terpene (mono- and ses-
quiterpene) emissions and primary number emissions show stronger correlation with largest CCN, but weak to non-existent
with smaller diameter CCN.

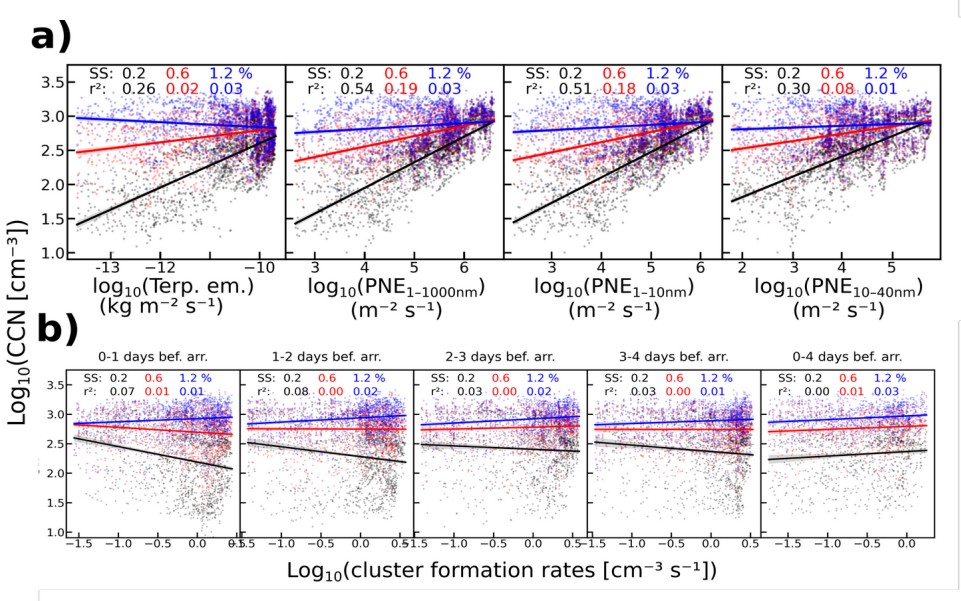

Figure 16: Panel a): Scatter plots of modelled CCN at the station and various model parameters for the last 4 days. Panel b)
Correlation of final [CCN] and cluster formation rates 0–1, 1–2, 2–3 and 3–4 days before the trajectory arrives at the sta-
tion.

## 4 Conclusions

The SOSAA-FP model framework has shown potential in modelling various atmospheric components and concentrations us-
ing global emissions datasets. While there are multiple steps that greatly simplify the three-dimensional transport of atmo-
spheric components, the comparison between observations and model results shows that the modelling scheme can give
valuable information about atmospheric aerosol and gas concentrations, composition and their sources. This allows the as-
sessment of the impact coming from sources of airmasses, emissions, meteorology, and seasons. The results seem especially
promising for bigger aerosol particles, for which the bulk particulate mass, composition and CCN activity was modelled
well, despite the model struggling more with number size distributions of smaller particles and some gas phase species. This





gives confidence in future use of SOSAA-FP for aerosol properties and phenomena, such as air quality, cloud formation and aerosol optics. Based on the sensitivity studies performed, the CCN concentrations are very sensitive to primary particle emissions, especially outside growing season. This highlights the importance of accurate estimation of size dependent primary particle emissions, as they have the most direct impact on CCN concentrations and the aerosol size distributions. The effect of the BVOC emissions on the CCN concentrations is comparable to that of primary aerosols during summertime

but diminished outside the growth season, as can be expected in such northern latitudes. While the winter months were left outside this study, it seems plausible to assume that the effect of PNE continues to be strong even during the periods when anthropogenic energy production and heating peaks and light conditions are unfavourable to BVOC emissions. Increased formation of atmospheric clusters had a negative impact on the smallest supersaturation CCN number concentration, but its effect became positive towards higher supersaturation classes. Even then, the effect was on average much smaller than the

changes in either BVOC or primary particle emissions, but showed a strong diurnal pattern where daytime responses of the smallest CCN sizes were comparable to those of the BVOC emissions. The ZERO-simulation analysis suggests that the effect of nucleation is proportional to the change in formation rate to the power of the response. As the responses are close to zero, the necessary change in cluster formation rates to make a difference in CCN concentrations needs to be orders of magnitudes instead of, say, doubling. As higher cluster formation rates usually occur in cleaner air when the condensation sink is

lower, CCN pathway through new particle formation seems to be suppressed whenever primary particle emissions are present, diminishing its effect on CCN concentrations. The sources of the airmasses during high SOA mass, usually meaning southerly or south-easterly origins in SMEAR II, generally did not coincide with the sources when new particle formation was strong, usually northerly or north-westerly origins. The modelled OA mass correlated with mean temperature along the trajectory, which agrees with previous studies based on local observations and trajectory analysis (e.g. Paasonen et al., 2013;

Yli-Juuti et al., 2021). The strong response to PNE is understandable considering how long it takes for nanometre sized particles to grow to even the smallest CCN sizes, whereas the emitted particles are already given a 'head start'. Furthermore, primary emissions increase the coagulation sink and greatly decrease the survivability of the nanoparticles. A notable result from the model is how increased cluster formation often lowers concentrations of the largest CCN, and in the model this could happen only via depletion of low volatility vapours from the otherwise growing Aitken mode particles. This is not to

say nucleation is not significant in creating CCN. While the model suggests that primary emissions dominate the aerosol dynamics, this may not always be the case in the future, and certainly not in the pre-industrial era. The ZeroPNE simulation showed that without primary aerosol emissions even the 0.4% supersaturation CCN would retain more than half of their concentrations, and the smallest CCN would be less affected. The largest CCN, however, would see a dramatic drop, although here we must point out that cloud processes and the related aqueous phase inorganic chemistry could potentially increase

their numbers from what was simulated here (e.g. Xavier et al., 2022). Another consideration is that the ZeroPNE was not a real pre-industrial simulation, as all other anthropogenic emissions, including $SO_2$ were kept as in BASE. Considering the ZeroNUC simulation, the drop in 0.4% CCN class was only approximately 15%, and even in the 1.2% CCN class the change





was approximately –40%. The sensitivity experiments imply a non-linear negative feedback dynamics in the CCN concentrations, where removing one process made the others increase their effect.

The model overestimated the particle concentrations in the 1–30 nm diameter range, but still underestimated the Aitken mode, suggesting that the growth of the newly formed particles is not correctly modelled. While the overshooting concentrations can be attributed to more frequent new particle formation in SOSAA than observed at SMEAR II, the missing Aitken mode particles would result from low survivability (high coagulation sink) and insufficient growth of the particles. When comparing the modelled and measured sulfuric acid and HOM concentrations, the model does not suffer from a considerable

lack of condensing vapours, even those that partition on the smallest particles. It is possible that the condensational growth alone is not able to reproduce the observed size distributions, at least with these saturation vapour pressures, but e.g. surface reactions and inorganic chemistry could play a significant role in enhancing SOA yields though mechanisms so far unaccounted for in the SOSAA model. The enhancing effect of antropogenic emissions on SOA production is well known generally (Riva et al., 2019) and that aromatic compounds (e.g. benzenes, polyaromatics and cycloalkanes) common in anthropo-

genic emissions markedly enhance production yields of highly oxidized molecules in the atmosphere through auto-oxidation reactions, and thus increase SOA mass (Rissanen, 2021). SOSAA-FP chemistry module in this work did not have autoxidation scheme from aromatics, meaning that some low volatility vapours are still missing from the model. Given how sluggishly the nucleation mode particles grow to CCN size, we should also not rule out that the gap in 30–300 nm modelled and observed number concentrations could simply come from underestimations in the particle emission dataset, further underlin-

ing the importance of proper estimations of primary number emissions in the CCN processes.

While this study did not focus on the meteorology, much of the dynamics and variation is strongly influenced by the meteorology. The airmass trajectories themselves are the result of atmospheric circulation, driven by the distribution of the solar energy. The environmental conditions and geographic locations along the trajectories set strong boundaries to which processes will dominate CCN formation. These constraints should be kept in mind when interpreting local observations and the pro-

cesses that led to them.

The key part of CCN formation from NPF is the efficient growth of the nucleation mode particles. During the development of this model and early experiments it became evident that just increasing BVOC emissions did not solve this conundrum, as the majority of low volatility vapours were accumulating on the larger particles, leading to overestimation compared to both aerosol mass and [HOM] measurements at SMEAR II, without significant shift of the nucleation mode towards Aitken and

CCN sizes. It seems that the vapour pressure based approach (which for extremely low volatility vapours and $H_2SO_4$ compounds in practice means condensation at kinetic limit) to gas-particle partitioning is not enough to reproduce the straightforward pathway of cluster formation to CCN in a detailed model such as SOSAA. Whether there is a process that either increases the survivability of the nucleation mode particles beyond what the Brownian coagulation model assumes, or if for example heterogenous chemistry on the nucleation mode particles increases their growth rates well beyond that calculated by

the Kelvin theory, is speculation, but something that should be investigated in more depth. Box models that utilize similar aerosol chemistry and physics have been evaluated in chamber experiments to the extent that doubting the modelling of co-

agulation seems unnecessary, and on the other hand the recent advances in heterogeneous chemistry of particles in the atmosphere give hope that these processes can be included in a general atmospheric model in the future. Meanwhile, these modelling results highlight the extremely dynamic and convoluted relation between atmospheric aerosol formation and primary emissions. While the more direct pathway from NPF to CCN could still be the main driver in remote locations, in the geographic region relevant to this study (Europe), anthropogenic influence is strong, despite the rural nature of the SMEAR II station itself. The results show that even with much improved air quality and emission control, human influence on the atmospheric fine particles is substantial.

## 5 Acknowledgements

PC extends his gratitude to the Helsinki Edinburgh program.

## 6 Code/Data availability

The processed model data used for figures is available at https://doi.org/10.5281/zenodo.14602389 (Clusius, 2025). SOSAA-FP code, and full model input and output is available at request from the corresponding author.

## 7 Author contribution

MB and PP outlined the study. PC, MB, CX, PZ, JT performed the model development and data preprocessing, PC conducted the simulations and evaluations, analyzed the simulation results, wrote the original paper and plotted all figures, under supervision of PP, MB and PP. MÄ, FG processed and provided ACSM data. BF, PP, TP and MK provided valuable comments and insights for the manuscript and model development. All authors contributed to the paper preparation, discussion, and writing.

## 8 Competing interests

The contact author has declared that none of the authors has any competing interests.

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
