# Peer review of "Modelling the impact of anthropogenic aerosols on the CCN concentrations in rural boreal forest environment"

_EGUsphere, 2025_

## Author Comment (AC1)

We would like to thank both reviewers for your constructive and valuable comments on our manuscript. I, as the corresponding author, would also like to apologize for the slow response to them on my part due to personal challenges with my work schedule this autumn. Please consider our edits in the text as improvements, and we hope you find our response adequately addressing the points raised.

In this document, and the manuscript, we have marked the additions to paper in red. Additionally, in the manuscript, where text has been moved in the document but essentially keeping the context intact and only accommodating for the grammar and context, the text is marked in blue.

In addition to addressing the specific reviewer comments, we have updated Figure 4 by adding a legend to panel (a) and corrected an error in panel (b) where the axes were in the wrong order, resulting in the least-squares-fit line being inverted. This error was only in the plotting and did not change the statistical figures.

The line numbers refer to the revised document unless otherwise noted.

Yours, Petri Clusius

**RC1: 'Comment on egusphere-2025-39', Anonymous Referee #1, 16 Jul 2025**

The study employs a novel Lagrangian modeling framework (SOSAA-FP) to investigate the impact of anthropogenic aerosols on cloud condensation nuclei (CCN) concentrations in a boreal forest environment. By combining global emission datasets, backward trajectories, and detailed aerosol dynamics, the work provides insights into the relative contributions of primary emissions and secondary aerosol formation to CCN, highlighting the importance of anthropogenic influences even in rural settings. This approach advances understanding of aerosol-cloud interactions and their implications for climate modeling. Yet several questions should be clarified before its publication.

**Major:**

1. Mechanisms Behind Weak Sensitivity to Cluster Formation Rates: The study finds that changes in atmospheric cluster formation rates have a relatively weak impact on CCN concentrations, which is unexpected given the importance of new particle formation (NPF) in aerosol dynamics. What specific mechanisms or feedbacks within the model could explain this weak sensitivity? For instance, how do the interactions between primary particles and newly formed clusters influence the overall CCN population? Understanding these mechanisms is crucial for refining models and improving predictions of aerosol impacts on climate.

To us also this result was surprising. Therefore we made sure that this was not an error in the code, but also by trying with significantly more stronger nucleation chemistries, such as those calculated with B3LYP/CBSB7//RICC2/aug-ccpV(T+d)Z level of theory (Olenius et al., 2013) level of theory that generally overpredict cluster stability and hence formation rates by orders of magnitude, and by including H2SO4-DMA nucleation (DMA emissions are unknown and a constant 1% factor of NH3 emissions were used, as is commonly done in literature). The consistent but very weak response in the CCN concentrations in all these tests suggested that the phenomenon is a persistent feature.

The reason for this weak response is the balance between the growth of the freshly formed particles and the coagulation sink. It should be noticed that the estimates of strong influence of NPF to CCN concentrations in modelling studies like Merikanto et al. (2009) represent the global sum of CCN concentrations. While NPF has a strong impact in marine environments, where other sources for CCN are weak, the contribution of NPF in continental air masses is much smaller. In terms of the observation based studies, e.g., at the SMEAR II station, the high values for the impact of NPF on CCN represent typically the situation when air masses are

coming from the clean north-western sector. When the air masses from eastern and southern sectors are considered, the influence of NPF becomes much smaller due to higher anthropogenic emissions both contributing to CCN production and causing higher condensation and coagulation sink suppressing the formation and survival of NPF particles to CCN sizes.

We can also estimate the formation efficiency of atmospheric clusters in ideal conditions where coagulation is the only sink term using the conversion function in Lehtinen et al. (2007). It calculates formation rates for particles of diameter  $D_{dia}$  when the coagulation sink and formation rate of 1 nm particles is known. Figure 1 shows the relative change of formation rates with 3 and 5 nm/h growth rate, using the median (measured) Mar–Oct 2018 coagulation sink at SMEAR II. This calculation shows why NPF does not necessarily produce significant amounts of CCN, particularly in the presence of particle emissions that increase the coagulation sink. The weak sensitivity therefore indicates the small contribution of NPF to CCN.

Figure 1. Formation rate  $J_{dia}$  as percentage of  $J_{lnm}$  with 3 and 5 nm/h constant growth rate using Lehtinen et al. (2007). The coagulation sink was taken from SMEAR II March-October median (shaded area shows 25-75% quantiles).

To clarify this point in the text, we have added the following to section 3.2 where sensitivity results are presented and discussed (line 567):

The weak impact of cluster formation to CCN concentration can be ultimately attributed to the low survival probability of 1–2 nm clusters and the resulting small contribution to the total CCN.

2. Role of Primary Emissions in CCN Formation: The results indicate that primary particle emissions play a substantial role in CCN concentrations even in a rural environment like SMEAR II. How do these findings reconcile with previous studies that emphasize the importance of secondary organic aerosols (SOA) in CCN formation? Specifically, what are the potential trade-offs between primary and secondary sources in different environmental conditions, and how might these findings influence the development of emission reduction strategies?

This is an interesting question. This study is the first modelling study where a detailed process model has been applied to airmasses that arrive at SMEAR II, taking into account the whole emission footprint (utilizing FLEXPART Source-Receptor Relationship). Other studies have used indirect methods such as PMF analysis from aerosol mass composition and trajectory analysis coupled with local observations. Compared with these studies, our work shows a much stronger effect of primary aerosol emissions, on average during the 8-month period 58% POA/OA, ranging from 17% in July to 95% in March. For example, (Heikkinen et al., 2021) reported only 1% and 9% POA / OA mass fraction in the months of July and March, respectively, in their 8-year ACSM study between 2012–2020. Their result is significantly lower than that of (Äijälä et al., 2019)

who reported on average 26% POA/OA fraction in campaigns conducted between 2008 and 2011, mostly taking place in the same months as were used in this work.

The distinction of primary emission in the model is simple in the sense that particles that are introduced into the model from the GAINS emissions are primary. However, GAINS treats organic aerosol mass that is formed immediately after emission from the emitted gaseous compounds as primary emissions, while it could still show as secondary in the analysis of ACSM signal. Our model also does not take into account the evaporation of primary particles (our POA is non-volatile), forming SVOCg that further oxidizes in gas phase and is then accumulating back on particles (Ciarelli et al., 2016), forming SOA (or some call it oxidized POA, OPOA). This could explain the high fraction of the primary particle in our result. The large difference in the POA/OA fraction that the two studies cited above understrike the uncertainties involved in the aerosol source appointment, and show that additional tools investigating them are welcome.

It is also possible that in the measured composition, some POA actually has undergone reactions in the particle phase and has become chemically indistinguishable from SOA as reported by Yazdani et al. (2023).

To improve the text we have added the following discussion, starting from line 479, where mass composition is discussed:

"Our work shows a much stronger effect of primary aerosol emissions than previous studies at SMEAR II. The average primary particle fraction of total OA (POA:OA) during the 8-month period was 58%, ranging from 17% in July to 95% in March. For example, Heikkinen et al. (2021) reported only 1% and 9% POA:OA mass fraction in the months of July and March, respectively, in their 8-years study between 2012 and 2020. Their result is significantly lower than that of Äijälä et al., (2019) who reported an average POA:OA fraction of 26% in campaigns conducted between 2008 and 2011, mostly in the same months as those used in this work. The discrepancy between our result and these studies can partly be explained by our treatment of primary particles as completely non-volatile, whereas they would likely evaporate some of their SVOC, which are then oxidized in the gas phase and eventually partitioning back to particles, showing as SOA. Ciarelli et al. (2016) used this approach in the CAMx model, which improved their model POA:OA fraction when compared to measurement, while increasing the underestimation of OA. This means that some SOA actually has sources in the primary particles, possibly masking the anthropogenic sources. It is also possible that in the measured composition, some POA actually has undergone reactions in the particle phase and has become chemically indistinguishable from SOA (Yazdani et al., 2023)."

The next paragraph is also new text (starting from 491), which was written in light of comment #4.

"On the other hand, the chemistry scheme used in this work does not include the autoxidation mechanism of aromatic compounds. During the study period the terpene emissions for the last 96 hours along the trajectory are substantially higher than aromatic emissions (on average 6 times higher and more than 8 times higher during summer). Assuming similar yields to monoterpenes, this would mean that including the aromatic autoxidation would increase the SOA mass (of mostly anthropogenic sources) on average by 10–15%, while in March, when the emissions are comparable to those of terpenes, the increase could be substantially higher."

3. Impact of BVOC Emissions on CCN Seasonality: The study demonstrates that BVOC emissions significantly affect CCN concentrations, especially during the growing season. However, the sensitivity of CCN to BVOC emissions is found to be highest in the largest CCN sizes. What are the implications of this finding for the role of BVOCs in cloud formation and climate feedback mechanisms? Additionally, how do seasonal variations in BVOC emissions influence the overall aerosol size distribution and CCN activity, and what are the potential feedbacks on regional climate?

While the climatological implications of the CCN and their sensitivity to the studied variables are of utmost importance, the scope of this paper was to model and quantify the effects of the different processes such as BVOC emissions to CCN concentrations at different supersaturations. It is difficult to even speculate on the (global feedback) effects of these changes due to the local perspective of this work. The implications of our

findings, especially the strong influence of primary emissions to CCN concentrations, and the suppression of NPF was already discussed in the conclusions, e.g. "Sensitivity experiments show that CCN concentrations are highly sensitive to primary particle emissions, especially outside the growing season".

In light of the comments we have received, we have edited Section 4 and made it more concise, as well as removed repetitive discussion, however the sentiment above is still in the conclusions.

The CCN concentrations at a given supersaturation are most sensitive to particle diameter, and while the 10-90% range of kappa values of CCN0.6% ranges from 0.9 to 0.42, the effect on critical diameter in the model ranges from 59 nm to 48 nm, respectively. Organic mass has lower activity and therefore the lower activity is mostly overcome by particle growth. This question has been investigated in measurements (e.g. Dusek et al, 2006), and it is still an interesting question that could be investigated in detail in future SOSAA-FP studies.

4. While the model shows reasonable agreement with observations for some components (e.g., organic aerosols and sulfates), significant discrepancies exist for others (e.g., nucleation-mode particles and gas-phase species like SO2 and H2SO4). The paper does not thoroughly address the potential sources of these biases, such as uncertainties in emission inventories, simplified deposition schemes, or missing chemical pathways (e.g., aromatic oxidation). A more rigorous discussion of model limitations and their impact on CCN predictions is needed.

We have added the following discussion to line 389 (new part in red): "The model overshoots [SO2] and consequently [H2SO4], even when the SO2 emissions were halved. This discrepancy is partly resulting from missing aqueous phase chemistry and the consequent sink term in the cloud droplets. The simple approach of halving the SO2 emissions to account for this leaves the actual variation in the SO2 sink unresolved. This is supported by the moderate positive correlations between the total precipitation along the trajectory and the model biases of SO2 and H2SO4 (r=0.33 and 0.30, respectively)."

Regarding aromatic autoxidation chemistry, such a scheme was not available when this study was conducted, but has been lately included in SOSAA-FP and has been used in a modelling study in Beijing, where it showed comparable yields to monoterpene autoxidation products. We added a paragraph which was discussed earlier on point #1.

Regarding emissions, we have used datasets that are publicly available, verified by peer reviewed publications, and are widely used by the scientific community. These emissions datasets are namely; Copernicus Atmosphere Monitoring Service; CAMS-GLOB-ANT, CAMS-GLOB-BIO, CAMS-GLOB-OCE, CAMS-GLOB-TEMPO, and for the particle number emissions ECLIPSE v5 project's GAINS emissions inventory. The main uncertainties in the emission datasets stem from its reliance on underlying statistical data and the methodologies used to process it, incomplete knowledge of the emission source, processes or their representativeness (activity data, emission factors, etc.), errors from spatial allocation (allocating emissions to specific locations and times), and structural differences such as sector definitions (SNAP, GFNR), and underlying assumptions between inventories. Uncertainties are a fundamental aspect of all emission inventories and are not unique to the one used in our study. Detailed information and their sources can be found both in main text (line 178) and in supplements (line 20-35). The major uncertainty or limitations of emissions were given in main text, section 3.1.1: "The biogenic emissions in CAMS are provided as monthly mean values, together with their mean daily profiles, and therefore are missing much of the day-to-day variation, and so will SOSAA.",

To study the sensitivity to the model input (and their uncertainties), we have selected all BVOC, primary particle emissions and nucleation rates. With the derived model sensitivity to these factors, the effect of the uncertainties can be assessed. To keep the paper in manageable size, we have not discussed the uncertainties of the input data in detail, as these have been extensively discussed in the dataset sources themselves.

Regarding deposition, particle dry deposition was discussed in Supplementary and was shown to align with numerous previous studies. The chemical dry deposition of any gas phase component was chosen to match that of average SO2, which, admittedly, is an oversimplification. This could lead to overestimation of gas phase NH3, which has higher water solubility (~10x Henry's law constant compared to SO2). The

implications of overestimating NH3 were already discussed in the paper in section 3.1.1. However the VOC deposition is unaccountable, and might be a significant source of uncertainty in most current air quality models (Liggio et al., 2025), and we have added the following sentence to line 240, where dry deposition is discussed: "This is an oversimplification especially with regards to VOCs, which are depositing with very varying rates, depending on their solubility and vapour pressures, leading to biases that could be substantial in most of the current air quality models (Liggio et al., 2025)."

The overestimation in the nucleation mode could come from 1) overestimating the formation of molecular clusters 2) insufficient growth of the nucleation mode 3) underestimating the losses such as (wet) deposition. The weak response of CCN concentrations to nucleation rates suggests that coagulation losses outweigh growth, pointing towards 2). This is supported by the underestimation of the CCN and Aitken mode and could result from missing AVOC autoxidation, incomplete oxidation chemistry, missing cloud processing and heterogeneous uptake of vapors. On the other hand, the overestimation in NH3 and H2SO4 concentrations would lead to 1), but since atmospheric cluster formation is most likely happening via other channels than pure H2SO4-NH3 nucleation, the overestimation could compensate for other missing sources. The revised section 4, "Discussion" discusses the possible reasons for the overestimation of the nucleation mode, especially considering the possible reasons for underestimated growth or survival probability.

**Minor:**

1. The text mentions the use of Welch's t-test and Mood's median test, but the results (e.g., p-values) are not included. Please provide the specific p-values or significance levels in the text to support the conclusions drawn from these tests.

Thank you for bringing this up – in the end these tests were only used to test whether measured spring, summer and autumn CCN concentrations showed significant differences from the median values with  $\alpha$ =0.01. Significance was only declared if p≤ $\alpha$ . We removed the mention of Welch's and Mood's test from 2.2.3 and added them in the location where it was only used (line 523, new text in red): "[...] only autumn concentrations differed significantly (–40 %, p<0.01, Mood's median test) from the total median", and section 2.2.3., line 345: To quantify model skill we use normalized mean bias factor (BNMBF, Eq. S3, Yu et al., 2006) and squared Pearson correlation coefficient ( $r^2$ ) to evaluate model performance. BNMBF is statistically robust, symmetric around zero (BNMBF = 0: no bias) and readily interpretable: the factor of under- or over-estimation is  $f = \left(1 + \left|B_{NMBF}\right|\right)^{sgn(B_{NMBF})}$  e.g. BNMBF = -0.5 means the model has a negative bias by a factor of (1+0.5)-1 = 1÷1.5, whereas BNMBF = 0.5 means bias (by overestimating) by a factor of 1.5.

2. The color scheme used in Figure 7 for different supersaturation levels is not very distinct. Please use a more contrasting color palette to enhance readability.

We increased the brightness of the red color, and changed the blue shading to light red. Also the scatter dots are smaller, in order to reduce clutter.

3. The reference list includes some inconsistencies in formatting (e.g., some references have full journal names while others are abbreviated). Please standardize the reference formatting to ensure consistency.

We have changed the journal names to abbreviated form (except "Science" and "Nature"), and corrected some errors in formatting such as missing subscripts.

4. Line 565: please change "backed with" to "supported by".

Correct, changed.

RC2: 'Comment on egusphere-2025-39', Anonymous Referee #2, 25 Aug 2025 reply

This study employs a novel Lagrangian modeling framework to investigate the origins and history of gas and aerosol components observed at the boreal forest measurement site, and their potential to act as CCN. I am impressed by the efforts the authors made for very detailed analysis and evaluation of the simulation results, with hourly, diurnal, and also seasonal variations. This study is methodologically rigorous, supported by extensive data, and provides significant scientific insights, particularly in understanding the relative contributions of anthropogenic and biogenic emissions to CCN formation. I recommend the publication of this work once the authors address my comments below.

Thank you for your encouraging, constructing and valuable comments on our manuscript.

**Major Comments:**

Page 10, Line 260-263, for the parameterizations estimating the volatility of organic compounds, as there are several other methods, such as Li et al., (2016) and Yang et al., (2023), I am curious why the authors picked the method of Stolzenburg et al. (2022)? Isaacman-Vanwertz and Aumont (2021) concluded that considering the average and distribution of error, the combined Daumit-Li method (modified to consider nitrates) represented a nearly optimal approach to estimating vapor pressure from a molecular formula. In addition, did the simulated organic compounds in the model in this study include CHOS compounds? If CHOS compounds were available, I recommend the authors re-examine whether Stolzenburg et al. (2022) could predict the volatility of compounds containing sulfur.

In the early stages of this work, we have used the model with saturation vapour pressures calculated with NANNOLAAL (Nannoolal et al., 2008) and EVAPORATION (Compernolle et al., 2011), which take into account the functional groups in the molecule and is therefore more advanced method than Stolzenburg et al. (2022, later referred as STOL). Of these NANNOLAAL shows on average lower vapor pressures. The method used in this work (STOL) agrees with Nannollaal in the lower end of the Psat spectrum, but shows lower psat on in the upper end. This has implications especially in the AVOC oxidation products (see figure below). While the difference in the modelled PM1 with STOL (+5%) when compared to NANNOLAAL was small, STOL produced stronger growth of the nucleation mode and thus increased the survivability of the sub-10nm particles. As we wanted to decrease the sources of underestimation of the growth of nucleation mode – crucial to the contribution of NPF to CCN formation – we chose the STOL method.

Figure 1: Saturation mixing rations from Stolzenburg et al. (2022) and Nannollaal. 1:1 line with red, shaded area shows mixing ratios <10ppb, representative relevant atmospheric mixing ratios of BVOC oxidation products.

Regarding organosulfates, the MCM chemistry contains only 27 organosulfates, which come from oxidation of DMS, and generally have very high vapour pressures. Only 4 out of the 27 compounds show saturation vapour pressures below 1 atm in -13°C when calculated with methods such as NANNOLLAAL, EVAPORATION and that of Myrdal and Yalkowsky (1997), and even their saturation vapor pressures are above 10-3 atm. Therefore, we do not even consider their condensation on the particles.

These compounds could still partition in the aqueous phase based on Henry's law, but as stated in the manuscript, this effect has not been taken into account in SOSAA. The two methods suggested above (Li et al., (2016) and Yang et al., (2023)) were not known to us at the time, but we will certainly consider them in future studies.

**Minor comments:**

(1) The manuscript is written very long. I recommend the authors could somehow shorten it and highlight the significance of this study. For Section 4, the Discussion and Conclusions, I recommend separate discussion and conclusions. Section 4 is too long and too detailed. The readers would like to have a more straight conclusion and significance/implication of this study.

We agree that the manuscript is already quite long, and the additional discussion included during the review process has further somewhat increased its length (hopefully also improving the clarity and completeness of the study). One reason for this is that the paper aims to transparently describe the method so that readers have as clear a view as possible of its strengths as well as its limitations. As the method has not been published

previously, we do not have the advantage that more established models offer, where the focus can be placed primarily on the results. Since this manuscript also seeks to provide insight into CCN formation processes rather than serving solely as a model description paper, the scope has inevitably become rather broad. We are therefore hesitant to move full sections to the Supplement, as condensing the text substantially would require a near-complete restructuring of the manuscript. However, we addressed this issue with two substantial edit:

- 1. In Section 3.3, the accompanying figures were moved to supplementary, and the text was edited accordingly. The paragraph starting "Figure 12 shows the geographical origins.." (line 599 in the old text) was removed, but its Fig. 12 was kept as S21, and reference to it was added to the previous paragraph (unchanged text in black, additions in red, line 631): "[...] coinciding with some of the highest CCN concentrations in the studied time period (see also Fig. S21)."
- 2. We have edited the Discussions and Conclusions, by splitting them in separate sections. We moved any discussion which was specific to a particular result to the respective place in the Results. This removed unnecessary repetition, while at the same time conveying the same information to the reader as in the first version in a more concise way. The remaining Discussion now concentrates on the most significant differences between modelled and observed particle size distributions, and how these could be addressed in future studies. In the revised version, the text that has been moved (with necessary grammatical changes) from Discussion and Conclusions to Results, is marked with blue, to distinguish it from completely new text, marked with red. As the changes are numerous and scattered, it would be impractical to list every one of them separately here. However, we have included the updated sections 4 and 5 at the end of this document.
- (2) In the discussion, the authors had emphasized the importance of the heterogeneous chemistry and particle-phase chemistry in CCN formation. Besides it, the phase state of organic aerosols may also play an important role in CCN formation pathways (Reid et al., 2018; Shiraiwa et al., 2017), and online simulation of organic aerosol phase state has been coupled in several chemical transport models (Rasool et al., 2021; Zhang et al., 2024) which could be applied to examine its effect in CCN. The potential effect of phase state on cluster formation or gas-particle partitioning is recommended to be discussed.

Thank you for bringing up this important point. The applied model employs temperature-dependent saturation vapour pressures to estimate the dynamic uptake and evaporation of organic compounds to and from the particle phase. The SOSAA model does not assume gas—particle partitioning equilibrium. As noted in the referenced studies, viscosity correlates strongly with saturation vapour pressure; therefore, temperature-induced changes in viscosity may be implicitly captured by the method used to estimate saturation vapour pressures. However, the saturation vapour pressures do not depend on ambient relative humidity, which, according to Rasool et al., was found to be a good predictor of viscosity (and hence volatility). We have added the following sentences to lines 462, where the potential reasons for the slower growth of the nucleation mode are discussed (new text in red):

"[...] although they might not be deposited in rainout in case of cloud droplet evaporation.

Finally, the 'particle phase' in the model is masking rather complex microphysics where the actual phase state can vary from liquid to glassy (Reid et al., 2018) depending on the environmental conditions, affecting their mixing state, water solubility, volatility and SOA uptake (Shiraiwa et al., 2017; Rasool et al., 2021; Zhang et al., 2024)."

(3) Page 25, Line 521, supersaturation classes of 0.4 % should be 0.6 % as Figure 9 showed.

Thank you, fixed.

**References (additions to the MS marked in red)**

Äijälä, M., Daellenbach, K. R., Canonaco, F., Heikkinen, L., Junninen, H., Petäjä, T., Kulmala, M., Prévôt, A. S. H., and Ehn, M.: Constructing a data-driven receptor model for organic and inorganic aerosol – a synthesis analysis of eight mass spectrometric data sets from a boreal forest site, Atmos. Chem. Phys., 19, 3645–3672, https://doi.org/10.5194/acp-19-3645-2019, 2019.

Ciarelli, G., Aksoyoglu, S., Crippa, M., Jimenez, J.-L., Nemitz, E., Sellegri, K., Äijälä, M., Carbone, S., Mohr, C., O'Dowd, C., Poulain, L., Baltensperger, U., and Prévôt, A. S. H.: Evaluation of European air quality modelled by CAMx including the volatility basis set scheme, Atmos. Chem. Phys., 16, 10313–10332, https://doi.org/10.5194/acp-16-10313-2016, 2016.

Compernolle, S., Ceulemans, K., and Müller, J.-F.: EVAPORATION: a new vapour pressure estimation methodfor organic molecules including non-additivity and intramolecular interactions, Atmospheric Chem. Phys., 11, 9431–9450, https://doi.org/10.5194/acp-11-9431-2011, 2011.

Olenius, T., Kupiainen-Määttä, O., Ortega, I. K., Kurtén, T., and Vehkamäki, H.: Free energy barrier in the growth of sulfuric acid–ammonia and sulfuric acid–dimethylamine clusters, J.Chem. Phys., 139, 084312, https://doi.org/10.1063/1.4819024, 2013.

Olenius, T., Pichelstorfer, L., Stolzenburg, D., Winkler, P. M., Lehtinen, K. E. J., and Riipinen, I.: Robust metric for quantifying the importance of stochastic effects on nanoparticle growth, Sci Rep, 8, 14160, https://doi.org/10.1038/s41598-018-32610-z, 2018.

Heikkinen, L., Äijälä, M., Daellenbach, K. R., Chen, G., Garmash, O., Aliaga, D., Graeffe, F., Räty, M., Luoma, K., Aalto, P., Kulmala, M., Petäjä, T., Worsnop, D., and Ehn, M.: Eight years of sub-micrometre organic aerosol composition data from the boreal forest characterized using a machine-learning approach, Atmos. Chem. Phys., 21, 10081–10109, https://doi.org/10.5194/acp-21-10081-2021, 2021.

Lehtinen, K. E. J., Dal Maso, M., Kulmala, M., and Kerminen, V.-M.: Estimating nucleation rates from apparent particle formation rates and vice versa: Revised formulation of the Kerminen–Kulmala equation, Journal of Aerosol Science, 38, 988–994, <a href="https://doi.org/10.1016/j.jaerosci.2007.06.009">https://doi.org/10.1016/j.jaerosci.2007.06.009</a>, 2007.

Liggio, J., Makar, P., Li, S.-M., Hayden, K., Darlington, A., Moussa, S., Wren, S., Staebler, R., Wentzell, J., Wheeler, M., Leithead, A., Mittermeier, R., Narayan, J., Wolde, M., Blanchard, D., Aherne, J., Kirk, J., Lee, C., Stroud, C., Zhang, J., Akingunola, A., Katal, A., Cheung, P., Ghahreman, R., Majdzadeh, M., He, M., Ditto, J., and Gentner, D. R.: Organic carbon dry deposition outpaces atmospheric processing with unaccounted implications for air quality and freshwater ecosystems, Sci. Adv., 11, eadr0259, https://doi.org/10.1126/sciadv.adr0259, 2025.

Myrdal, P. B. and Yalkowsky, S. H.: Estimating Pure Component Vapor Pressures of Complex Organic Molecules, Ind. Eng. Chem. Res., 36, 2494–2499, https://doi.org/10.1021/ie9502421, 1997.

Nannoolal, Y., Rarey, J., and Ramjugernath, D.: Estimation of pure component properties, Fluid Phase Equilibria, 269, 117–133, https://doi.org/10.1016/j.fluid.2008.04.020, 2008.

Yazdani, A., Takahama, S., Kodros, J. K., Paglione, M., Masiol, M., Squizzato, S., Florou, K., Kaltsonoudis, C., Jorga, S. D., Pandis, S. N., and Nenes, A.: Chemical evolution of primary and secondary biomass burning aerosols during daytime and nighttime, Atmos. Chem. Phys., 23, 7461–7477, <a href="https://doi.org/10.5194/acp-23-7461-2023">https://doi.org/10.5194/acp-23-7461-2023</a>, 2023.

**Revised Discussion and Conclusions:**

**4 Discussion**

Although the model evaluation demonstrated promising results, challenges remain. The model SOSAA overestimated the particle concentrations within the 1–30 nm diameter range while underestimating the Aitken mode. Paasonen et al. (2016) estimated that the applied PNE in nucleation mode is clearly underestimated, a conclusion subsequently corroborated by observations in Beijing (Kontkanen et al., 2020). Overestimating the concentrations while very likely underestimating the emissions in this size range would suggest that the growth of the newly formed particles is not sufficiently efficient in the model. The overestimated nucleation mode concentrations may be partly attributable to more frequent new particle formation events in SOSAA compared to observations at the SMEAR II station, but the deficit in Aitken mode particles likely stems from low particle survivability due to coagulation sink, and/or insufficient particle growth.

Our simulations showed that just increasing the current BVOC emissions did not sufficiently favor this growth as the majority of low volatility vapours were accumulating on the larger particles, leading to overestimation in both aerosol mass and [HOM] measurements at SMEAR II, without significant shift of the nucleation mode towards Aitken and CCN sizes. When comparing the modelled and measured sulfuric acid and HOM concentrations, the model does not suffer from a considerable lack of low volatility vapours. SOSAA-FP chemistry module in this work did not have an autoxidation scheme for aromatics, meaning that some low volatility vapours are still missing from the model.

The capacity of AVOC emissions to enhance SOA production is well-established (Riva et al., 2019). Specifically, aromatic compounds such as benzenes, polyaromatics, and cycloalkanes, which are prevalent in anthropogenic emissions, significantly increase the yield of highly oxidized molecules in the atmosphere via autoxidation reactions, thereby increase SOA mass (Rissanen, 2021). Furthermore, it is possible that condensational growth alone is insufficient to replicate the observed size distributions, at least within the constraints of the current estimations of saturation vapour pressures. Other processes, such as surface reactions and inorganic particle phase chemistry, and mechanisms not yet represented in the SOSAA model may contribute significantly to enhancement of SOA yields. This suggests that the model's vapour pressure-based approach to gas-particle partitioning, which for extremely low volatility compounds and H2SO4 effectively leads to condensation at the kinetic limit, is inadequate for accurately simulating the pathway from cluster formation to CCN. Including processes that enhance the growth of the nucleation mode particles, such as cloud processing or heterogeneous chemistry, or increase their survivability, such as primary particle evaporation (discussed in section 3.1.2) could improve the model results, and warrants further in-depth investigation. An additional mechanism to consider is the impact of stochastic growth, which Olenius et al. (2018) demonstrated provides a more accurate representation of growth for particles below approximately 5 nm in diameter. Box models utilizing similar aerosol physics mechanisms as was used in this study have been evaluated in chamber experiments to the extent that doubting the modelling of coagulation seems unnecessary. Concurrently, recent advances in understanding the heterogeneous chemistry of atmospheric particles provide a solid foundation for integrating these processes into future models. Lastly, given the characteristically slow growth of nucleation mode particles to CCN sizes, the discrepancy in the

30–300 nm modelled and observed number concentrations may also originate from an underestimation within the particle emission dataset. This possibility underscores the critical importance of accurate primary particle number emission estimates for modelling CCN processes.

**5 Conclusions**

In this work we have used a novel Lagrangian modelling framework to examine the origins and history of gas and aerosol components observed at the boreal forest measurements site SMEAR II. The SOSAA-FP framework integrates global emission datasets, FLEXPART backward trajectories, and the detailed atmospheric chemistry and aerosol dynamics of the SOSAA model. The period from March to October 2018 was simulated by using this framework with one-hour temporal resolution. The SOSAA-FP framework simplifies three-dimensional transport of atmospheric components while providing a more comprehensive description of gas-phase chemistry, aerosol dynamics and composition than large-scale 3-D models. Model evaluation against observations confirmed the framework's ability in assessing the impacts of airmass origins, emissions, meteorology, and seasonal variations. The model performed particularly well for larger aerosol particles, agreeing with measured bulk particulate mass, composition, and CCN concentrations. However, challenges persist in reproducing the size distributions of smaller particles and certain gas-phase species, and these issues are partly attributable to uncertainties in the input data. Despite these limitations, the SOSAA-FP framework demonstrates significant potential for future applications in air quality, cloud formation, and aerosol optical properties. Our results from the study period showed large temporal variation in the impacts of primary aerosol emissions to CCN concentrations at SMEAR II station. On average between March and October 2018, the modelled [CCN0.4%] and [CCN1.2%] without primary particle emissions decreased by 56% and 33% and without cluster formation by 22% and 48%, respectively. These results illustrate a non-linear, compensatory relationship between the two sources: the deficit in CCN in the ZeroPNE from missing primary emissions is partially offset by enhanced new particle formation. This effect arises primarily from reduced coagulation sink for molecular clusters, thereby improving their survivability, and the shift in the condensation sink towards smaller sizes, increasing growth rates of smaller particles.

Sensitivity simulations show that CCN concentrations strongly respond to primary particle emissions, especially outside the growth season, underlining the importance of size-dependent primary particle emission data for models. These modelling results highlight the highly dynamic and complex relation between atmospheric aerosol formation and primary emissions. Although the direct pathway from NPF to CCN may dominate in remote regions, the geographic domain of this study (Northern Europe) is subject to considerable anthropogenic influence, despite the rural setting of the SMEAR II station. These findings suggest that, despite decades of improved air quality and emission control strategies, anthropogenic activities continue to exert a substantial influence on atmospheric fine particle concentrations.

**References from Rev. #2:**

- 1. Isaacman-VanWertz, G. and Aumont, B.: Impact of organic molecular structure on the estimation of atmospherically relevant physicochemical parameters, Atmos. Chem. Phys., 21, 6541-6563, 10.5194/acp-21-6541-2021, 2021.
- 2. Li, Y., Pöschl, U., and Shiraiwa, M.: Molecular corridors and parameterizations of volatility in the chemical evolution of organic aerosols, Atmos. Chem. Phys., 16, 3327-3344, 10.5194/acp-16-3327-2016, 2016.
- 3. Rasool, Q. Z., Shrivastava, M., Octaviani, M., Zhao, B., Gaudet, B., and Liu, Y.: Modeling Volatility-Based Aerosol Phase State Predictions in the Amazon Rainforest, ACS Earth and Space Chemistry, 5, 2910-2924, 10.1021/acsearthspacechem.1c00255, 2021.
- 4. Reid, J. P., Bertram, A. K., Topping, D. O., Laskin, A., Martin, S. T., Petters, M. D., Pope, F. D., and Rovelli, G.: The viscosity of atmospherically relevant organic particles, Nat. Commun., 9, 956, 10.1038/s41467-018-03027-z, 2018.
- 5. Shiraiwa, M., Li, Y., Tsimpidi, A. P., Karydis, V. A., Berkemeier, T., Pandis, S. N., Lelieveld, J., Koop, T., and Pöschl, U.: Global distribution of particle phase state in atmospheric secondary organic aerosols, Nat. Commun., 8, 15002, 10.1038/ncomms15002, 2017.
- 6. Yang, X., Ren, S., Wang, Y., Yang, G., Li, Y., Li, C., Wang, L., Yao, L., and Wang, L.: Volatility Parametrization of Low-Volatile Components of Ambient Organic Aerosols Based on Molecular Formulas, Environmental Science & Technology, 57, 11595-11604, 10.1021/acs.est.3c02073, 2023.
- 7. Zhang, Z., Li, Y., Ran, H., An, J., Qu, Y., Zhou, W., Xu, W., Hu, W., Xie, H., Wang, Z., Sun, Y., and Shiraiwa, M.: Simulated phase state and viscosity of secondary organic aerosols over China, Atmos. Chem. Phys., 24, 4809-4826, 10.5194/acp-24-4809-2024, 2024.